# Comprehensive Investigation of Promising Techniques to Enhance the Voltage Sharing among SiC MOSFET Strings, Supported by Experimental and Simulation Validations

**Weichuan Zhao** [1,*]**, Sohrab Ghafoor** [1]**, Gijs Willem Lagerweij** [2]**, Gert Rietveld** [3,4]**, Peter Vaessen** [1,5] **and Mohamad Ghaffarian Niasar** [1,*]

1   EWI-HVT Group, Delft University of Technology, Mekelweg 4, 2628 CD Delft, The Netherlands; s.ghafoor-1@tudelft.nl (S.G.); p.t.m.vaessen@tudelft.nl (P.V.)
2   Prodrive Technologies, Science Park Eindhoven 5501, 5692 EM Son, The Netherlands; gijs.lagerweij@prodrive-technologies.com
3   Power Electronics Group, University of Twente, Drienerlolaan 5, 7522 NB Enschede, The Netherlands; grietveld@vsl.nl
4   VSL National Metrology Institute, Thijsseweg 11, 2629 JA Delft, The Netherlands
5   KEMA Laboratories, Klingelbeekseweg 195, 6812 DE Arnhem, The Netherlands
*   Correspondence: w.zhao-3@tudelft.nl (W.Z.); m.ghaffarianniasar@tudelft.nl (M.G.N.)

**Abstract:** This paper comprehensively reviews several techniques that address the static and dynamic voltage balancing of series-connected MOSFETs. The effectiveness of these techniques was validated through simulations and experiments. Dynamic voltage-balancing techniques include gate signal delay adjustment methods, passive snubbers, passive clamping circuits, and hybrid solutions. Based on the experimental results, the advantages and disadvantages of each technique are investigated. Combining the gate-balancing core method with an RC snubber, which has proven both technically and commercially attractive, provides a robust solution. If the components are sorted and binned, voltage-balancing techniques may not be necessary, further enhancing the commercial viability of series-connected MOSFETs. An investigation of gate driver topologies yields one crucial conclusion: magnetically isolated gate drivers offer a simple and cost-effective solution for high-frequency (HF) applications (2.5–50 kHz) above 8 kV with an increased number of series devices. Below 8 kV, it is advantageous to move the isolation barrier from the gate drive IC to an optocoupler and isolated supply, allowing for a simple design with commercially available components.

**Keywords:** dynamic voltage balancing; series-connected MOSFETs; gate-balancing core method; improved RC snubber; Zener clamping; magnetically isolated gate driver; high-voltage switch; MOSFET string

## 1. Introduction

As the integration of renewable energy sources into the electricity grid continues to increase, the deployment of medium-voltage (MV) high-power converters, modular multilevel converters (MMCs) [1,2], and solid-state transformers (SSTs) [3,4] will become crucial in the near future. For the next generation of these power-electronic (PE)-based systems, high efficiency, high voltage (HV), and high power are essential for a wide range of applications, as summarized in Figure 1. In this context, the role of semiconductor devices with the capability to operate at high switching frequencies, withstand elevated temperatures, and exhibit low switching losses becomes even more important.

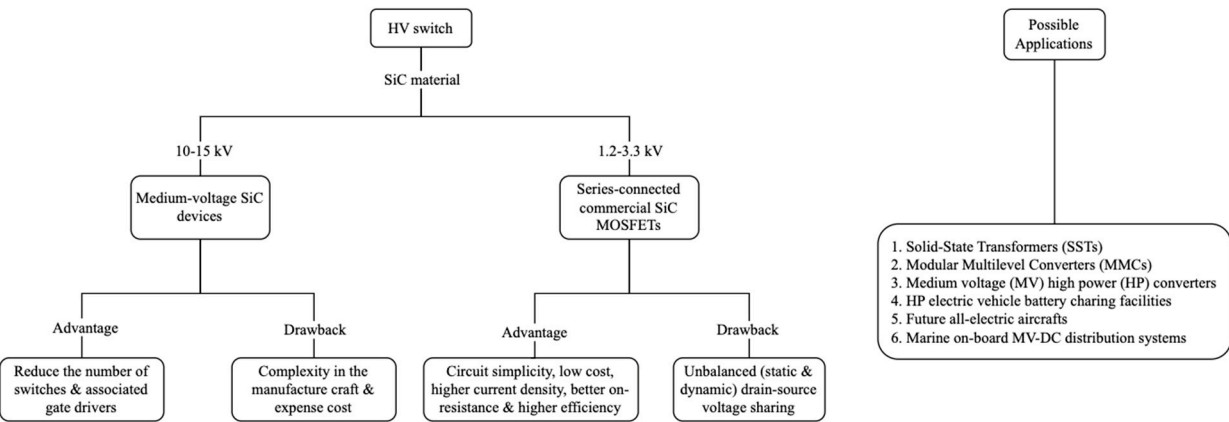

**Figure 1.** Properties for different types of HV switches (**left**) and their possible applications (**right**) [1–18].

Compared to silicon (Si), silicon carbide (SiC) excels in several properties, such as reasonable electron mobility, higher critical field strength, and higher thermal conductivity. SiC MOSFETs may have lower on-resistance $R_{DS(on)}$, higher blocking voltage, and higher operational temperature, and can be used at higher switching frequencies [19]. Compared to IGBTs, SiC MOSFETs have no tail-current characteristics during the turn-off period, which contributes to lower switching loss and shorter turn-off delay.

Currently, MV SiC devices with a blocking voltage of 10–15 kV have sparked significant interest for use in HV applications, although they have not yet been commercialized [20,21]. The main limitations are the cost and the complexity of device manufacturing and packaging, which currently restrict the commercially available SiC MOSFETs to a maximum blocking voltage of 3.3 kV [22]. However, to overcome this blocking voltage limitation and achieve higher system voltages, it is possible to connect multiple commercially available SiC MOSFETs in series. Figure 1 provides a detailed overview of the advantages and drawbacks of such SiC MOSFET strings and MV SiC MOSFETs.

Using multiple low-cost and commercially available SiC MOSFETs in series allows for the required HV operation with simple yet mature gate-driving techniques. The major obstacle to implementing the series-connected operation is maintaining an equal voltage distribution across the SiC MOSFET string. This voltage balancing is essential to achieve equal stressing of the MOSFETs, and thus high reliability and higher operating voltages by reducing the required voltage derating. The voltage imbalance is primarily caused by mismatched gate driving signals, the existence of SiC MOSFET intrinsic (associated with its physical construction) and external parasitic capacitances and inductances [23], and the variation in SiC MOSFET off-resistance. These factors result in switching delays or time shifts (several ns to tens of ns), variation in $dV_{GS}/dt$, and voltage imbalance across the MOSFET string.

### 1.1. Factors Influencing the Drain-Source Voltage Distribution

The factors that can cause unequal drain-source voltage sharing of the SiC MOSFET string are elaborated in the following. A basic testing circuit composed of two series-connected SiC MOSFETs and one current-limiting resistor is proposed as shown in Figure 2.

**Variation in MOSFET switching delay** $\delta t_d$: According to the experimental results, ignoring the other possible factors, the intrinsic variation in the switching delay is not the dominant factor causing $V_{DS}$ imbalance. Furthermore, part of the variation in $t_{d(on)}$ and $t_{d(off)}$ between the measured values and the typical values from datasheets may be due to different measuring conditions. Generally, the variation in MOSFET parasitic parameters is limited if the devices are bought from good manufacturers and selected from the same production batch. However, the gate threshold voltage of SiC MOSFET may vary with temperature, and from device to device. This can contribute to a larger $\delta t_d$.

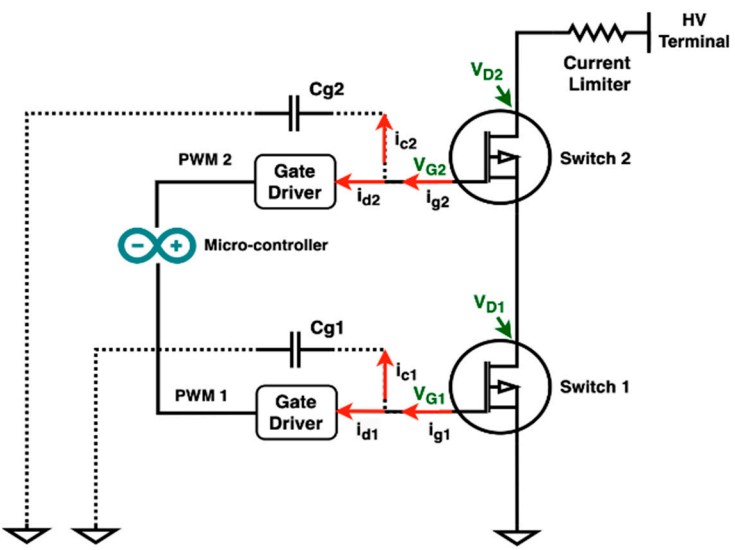

**Figure 2.** Basic schematic for testing two series-connected SiC MOSFETs.

Fifteen measurements of switch type IMW120R220M1H (Infineon) or C3M0280090D (Wolfspeed) were performed based on [24] and the median was taken as the typical value of $t_{d(on)}$ and $t_{d(off)}$. Among the obtained data, presented in Table 1, the intrinsic variations in $t_{d(on)}$ and $t_{d(off)}$ are 2.0 ns and 2.4 ns (IMW120R220M1H), and 2.3 ns and 1.6 ns (C3M0280090D). Therefore, the intrinsic variation $\delta t_d$ is much smaller than the other factors contributing to voltage imbalance.

**Table 1.** The turn-on and -off delay of two types of commercially available SiC MOSFETs.

| SiC MOSFET | Datasheet | | Measurement | | | |
|---|---|---|---|---|---|---|
| | $t_{d(on)}$ | $t_{d(off)}$ | $t_{d(on)}$ | $\delta t_{d(on)}$ | $t_{d(off)}$ | $\delta t_{d(off)}$ |
| IMW120R220M1H | 5.0 ns | 10.0 ns | 6.8 ns | $\pm 2.0$ ns | 12.8 ns | $\pm 2.4$ ns |
| C3M0280090D | 5.3 ns | 8.5 ns | 7.7 ns | $\pm 2.3$ ns | 11.6 ns | $\pm 1.6$ ns |

**Variation in gate driver switching delay $\delta t_d$:** The variation in switching delay that arises due to variations in the gate driver circuitry, components, and layout (i.e., the external factors) are much more significant than the intrinsic variation of the MOSFET. For example, the maximum switching delay variation $\delta t_{d(max)}$ reported in [25,26] of the isolated gate driver type STGAP2SICSN (e.g.) and that of the non-isolated gate driver type IXDD630MCI (e.g.) is around 40 ns. Moreover, according to [27], that of the opto-coupler type FOD3182 (e.g.) can even reach 160 ns. The value of $\delta t_{d(max)}$ can be reduced by sorting, binning, and matching components, thus reducing the voltage imbalance.

**Parasitic capacitances from MOSFET gates to ground:** Based on [28], as seen in Figure 2, the $C_{gi}$ existing from respective gates to ground is observed as the dominating factor contributing to the unbalanced $V_{DS}$ sharing among the MOSFET string. If the MOSFETs and driving components were identical, identical sink currents would flow to the drivers ($i_{d1} = i_{d2}$). However, due to the difference between the voltages $V_{Gi}$ with respect to ground, a variation in the gate voltage slopes occurs, which results in a different magnitude of capacitive currents $i_{ci}$ and creates a difference in total gate currents $i_{gi}$ leading to voltage imbalance.

In Figure 2, the total gate current $i_{gi}$ is the sum of the gate sink current $i_{di}$ and capacitive current $i_{ci}$ from the gate to ground. The $dv/dt$ at the gate of the top MOSFET is equal to that at the drain of the bottom MOSFET ($v_{G2} \approx v_{D1}$), while the gate of the bottom MOSFET is almost at ground potential. Thus, $dv_{G2}/dt$ is higher than $dv_{G1}/dt$ and the resultant capacitive current $i_{c2}$ is also larger than $i_{c1}$. Moreover, the difference in $C_{gi}$ from the stacked gate terminals to ground will also play a role in altering the intensity of the total

gate current $i_g$ when the number of the involved MOSFETs is large. For a small number of MOSFETs, the difference in $C_{gi}$ can be neglected. This shows that even with perfectly matched gate drivers, $i_{g2} > i_{g1}$, resulting in faster turn-off of the top MOSFET.

**Variation in MOSFET off-resistance** $R_{DS(off)}$: Due to the presence of the variation in SiC MOSFET $R_{DS(off)}$, the static $V_{DS}$ sharing of series-connected MOSFETs may be unbalanced. Balancing resistors ought to be connected in parallel with the SiC MOSFETs to achieve balanced static voltage sharing. Furthermore, during the $V_{DS}$ measurement, the impedance of the differential probes will influence the balancing resistor network, and thus should also be considered.

### 1.2. State-of-the-Art Solutions

Over the past decades, various solutions have been proposed to improve the voltage sharing between series-connected MOSFETs. These methods can be categorized into static and dynamic balancing. Most challenges are encountered in the dynamic balancing. Some of the proposed solutions are Zener clamping circuits, passive snubber circuits, and gate signal delay adjustment methods.

To avoid the SiC MOSFET breakdown caused by the unbalanced voltage sharing, the Zener clamping circuit is introduced in [29] and evaluated in [30]. The overvoltage across the MOSFET is eliminated by clamping $V_{DG}$ through the series-connected Zener diodes whose equivalent reverse breakdown voltage $V_Z$ is chosen based on the blocking voltage of the MOSFET. In [31], four types of passive snubbers are summarized with thorough principle elaboration. The purpose of introducing these passive snubbers is either to reduce the rise slew rate $dV_{DS}/dt$ or to clamp the $V_{DS}$ in such a way that the series-connected MOSFETs can share identical voltages.

The gate signal delay adjustment methods can compensate for the delay time variation without slowing down the switching speed. In [32], Kiyoaki reports an important technique named the gate-balancing core (GBC) method which uses gate-coupled magnetic cores to synchronize the SiC MOSFET gate drive currents. In [33], an improved RC snubber method is proposed by S. Chen, which has a combination of the passive snubbers and the gate signal delay adjustment methods. The key point of this method is the use of a three-port inductor whose primary windings are coupled within two snubber circuits and whose secondary winding is in series with the gate. The induced voltages from the secondary windings will be added back to the gate circuits to tune $V_{GS}$ and achieve identical gate currents.

Reliable and robust voltage-balancing techniques must achieve effective voltage balancing, minimize the number of components within the balancing circuit, simplify the gate-side control circuits, and introduce minimal switching losses [34]. The purpose of this paper is to provide a detailed and concise evaluation of various approaches dealing with the unbalanced voltage sharing among the SiC MOSFET string. The advantages and disadvantages of each method are discussed and the suitable conditions for the application of each method are provided.

### 1.3. Outline

In Section 2, methods for improving static $V_{DS}$ sharing of series-connected MOSFETs are introduced. Section 3 describes how the GBC method improves the dynamic $V_{DS}$ sharing of the MOSFET string in case a large variation in turn-off delay exists. In Section 4, two types of improved RC snubber methods are discussed. The output performances of these methods are evaluated and compared in a scenario with a large turn-off delay variation. Section 5 evaluates how the optimized Zener clamping method influences the $V_{DS}$ sharing of the MOSFET string in the presence of large turn-off delay variation and parasitic capacitances. In Sections 6 and 7, two types of gate drivers are described to achieve HV operation of the MOSFET string. Finally, in Section 8, the described methods are compared, and recommendations are given based on the application scenario.

## 2. Static Voltage-Balancing Method

The factors that can lead to the static voltage imbalance are thoroughly examined, underscoring the significance of this issue. A promising solution is then proposed, improving the static voltage sharing of the SiC MOSFET string. Additionally, the correct use of differential probes to measure the SiC MOSFET drain-source voltage is detailed, which is an important aspect that can significantly influence the static voltage sharing of the MOSFET string while performing drain-source voltage measurements.

As seen in Figure 3 (left), an unbalanced static voltage sharing issue occurs on the three series-connected MOSFETs. This phenomenon can occur due to variation in the off-resistance $R_{DS(off)}$ of the MOSFETs or the impact of the measurement probe. This issue can be effectively addressed through the utilization of static balancing resistors $R_{st}$, which equalizes the voltage stress in the series-connected devices at the cost of increased static power dissipation. The MOSFETs chosen for experiments are of type IMW120R220M1H. From its datasheet, if the applied drain-source voltage is 1.2 kV, the drain leakage current $I_{DSS}$ varies over two decades (0.2 µA to 95 µA) at an ambient temperature of 25 °C. Therefore, the corresponding MOSFET off-resistance $R_{DS(off)}$ varies from 6 GΩ to 12.6 MΩ.

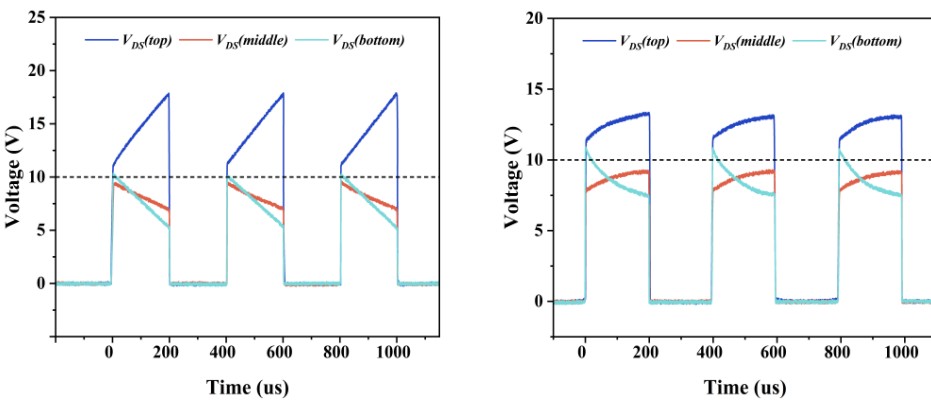

**Figure 3.** Measured unbalanced $V_{DS}$ sharing of the SiC MOSFET string without (**left**) and with (**right**) the identical balancing resistors (500 kΩ).

Generally, the value of the parallel balancing resistor $R_{st}$ should be selected to be at least 10 times smaller than the minimum $R_{DS(off)}$. The parallel combination of $R_{st}$ and $R_{DS(off)}$ will then be dominated by the balancing resistor, reducing the effect of variation in $R_{DS(off)}$. The power loss in each balancing resistor is calculated using (1). With a derating of 40% and a maximum dissipation of 1 W, a minimum $R_{st}$ of 500 kΩ is calculated.

$$P_{st} = \frac{U_{max}^2}{R_{st}} = \frac{(0.6 * V_{DSS})^2}{R_{st}} = \frac{(0.6 * 1200)^2}{500 * 10^3} \approx 1 \text{ W} \tag{1}$$

During the measurement of MOSFET drain-source voltages, even though the static balancing resistors ($R_{st} = 500$ kΩ) are applied, the static voltages along the SiC MOSFET string are not yet balanced, as shown in Figure 3 (right).

The reason for this phenomenon is that the impedance of the differential probes $R_r = 4$ MΩ and $R_b = 4$ MΩ must be considered as part of the balancing resistor network, as shown in Figure 4 (left), which influences the static voltage sharing. Hence, while observing the drain-source voltage, $R_{st}$ should be fine-tuned only during the tests to prevent static voltage imbalance. Figure 4 (right) shows the simplification of the resistive network during $V_{DS}$ measurement. Assuming the value of $R_{st(3)}$ that is applied across the bottom MOSFET is 500 kΩ, the presence of the probe impedance $R_{diff(23)}$ changes the required value of $R_{st(2)}$ for the middle switch to 400 kΩ, using (2). Similarly, the total equivalent resistance for the bottom two MOSFETs combined with $R_{diff(22)}$ and $R_{diff(23)}$ is about 571.4 kΩ. Therefore, the magnitude of $R_{st(1)}$ should be half the value of the obtained total equivalent resistance, calculated using (3).

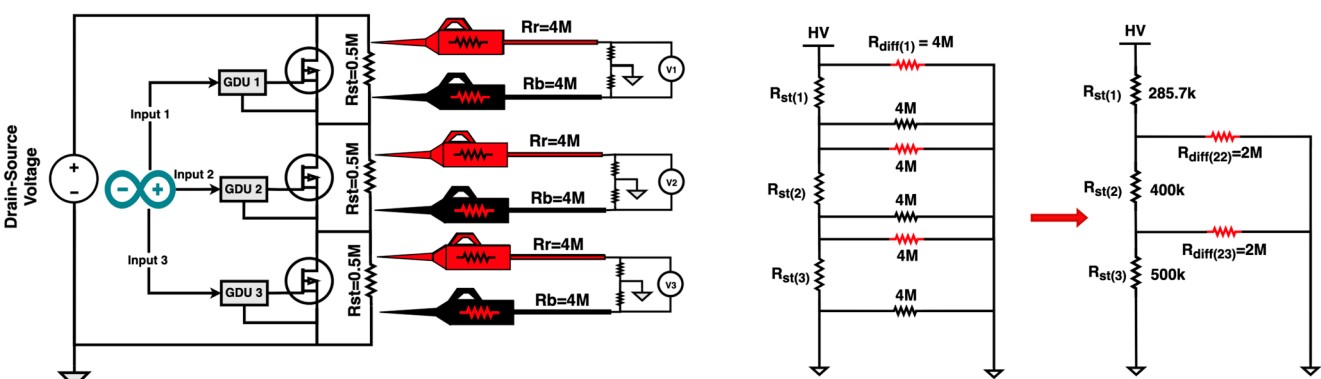

**Figure 4.** Schematic of the $V_{DS}$ measurement of the three series-connected MOSFETs (**left**) and the corresponding resistance ladder network (**right**).

$$R_{st(2)} = R_{st(3)} \big\| R_{diff(23)} = 400 \text{ k}\Omega \tag{2}$$

$$R_{st(1)} = \frac{1}{2} \Big[ \big( 2R_{st(2)} \big) \big\| R_{diff(22)} \Big] = 285.7 \text{ k}\Omega \tag{3}$$

After the application of the tuned static balancing resistors as shown in Figure 4 (right), the balanced static voltage sharing of the MOSFET string can be achieved (Figure 5). However, due to the existence of the parasitic capacitances and inductances, the unmatched MOSFETs, and gate drivers, balanced dynamic voltage sharing is not yet achieved.

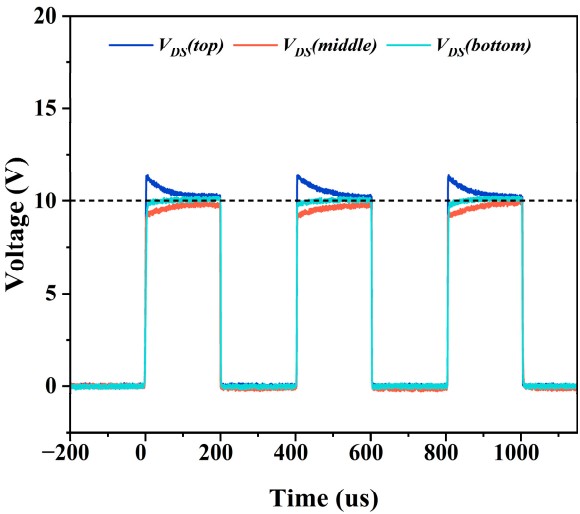

**Figure 5.** Measured overall $V_{DS}$ sharing of the MOSFET string with tuned balancing resistors.

Indeed, the balancing resistors with tuned values are only used to observe the statically balanced $V_{DS}$ waveforms on the oscilloscope during experiments. If the SiC MOSFET string is used under HV applications, the value of the required balancing resistors should be identical since there is no need to measure the drain-source voltages during normal operation.

## 3. Gate-Balancing Core (GBC) Method

This section evaluates the GBC method to improve the dynamic $V_{DS}$ sharing in a string of series-connected MOSFETs. This method is based on gate signal delay adjustment. The method is validated by experiments, which are relevant for a SiC MOSFET string with a considerable variation in the turn-off delay $t_{d(off)}$. In [32,35], the GBC method is proposed to synchronize mismatched gate currents. The key point of this method is the use of a coupled inductor with a high coupling factor *k*, which is well-coupled within the adjacent gate circuits of the SiC MOSFET string. The magnetic coupling will impose almost

identical gate sink currents ($I_{g1} \approx I_{g2}$), even if a considerable turn-off delay time variation $\delta t_{d(off)}$ exists.

When a slight turn-off delay $\delta t_{d(off)}$ is present in the bottom MOSFET, as depicted in Figure 6, the top MOSFET switches off faster. Its gate current $I_{g1}$ flows through the upper winding of the coupled inductor and returns to the gate driver. The induced current $I_{g2}$ occurs simultaneously with $I_{g1}$ on the lower winding through magnetic coupling. If the turns ratio is 1:1 and $k$ is close to 1, the magnitudes of the induced gate current $I_{g2}$ and the initial sink current $I_{g1}$ are identical. This synchronization of the gate currents leads to balanced voltage sharing among the series-connected MOSFETs. Figure 7 further illustrates the extension of the GBC method to a higher number of series-connected SiC MOSFETs. The inter-winding insulation requirements are relaxed since inductors are only coupled between consecutive MOSFETs.

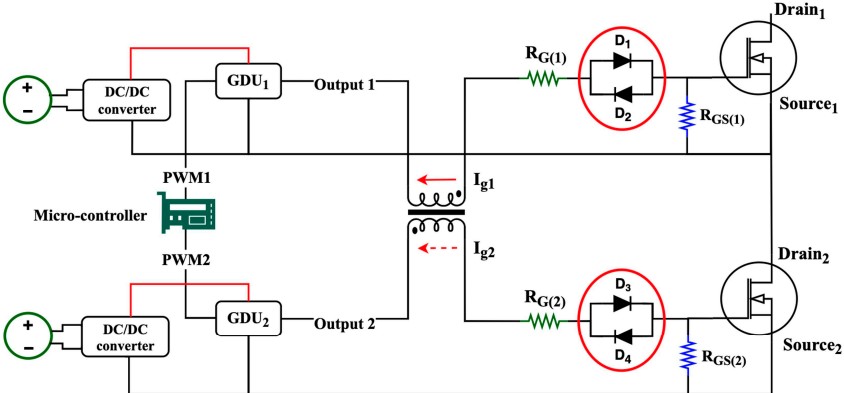

**Figure 6.** Schematic of the two series-connected SiC MOSFETs using the GBC method.

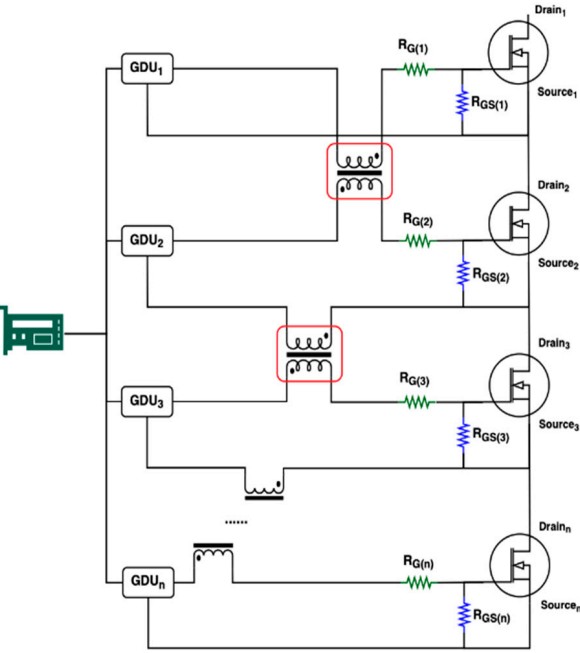

**Figure 7.** Schematic of the multiple series-connected SiC MOSFETs using the GBC method.

The coupled inductor leakage inductance $L_k$ can resonate with the dynamic input capacitance $C_{iss}$ of the MOSFET. It should be minimized to avoid excessive ringing or oscillations on the transient parts of the gate pulses, which influences the output performance of the MOSFET string. Thus, interleaved inductor construction is recommended. Indeed, (4) and (6) are derived to calculate the required magnetizing inductance $L_m$ and

leakage inductance $L_k$ in case of a particular variation $\delta t_{d(off)}$ among the series-connected MOSFETs [32,35].

In (4), $\delta t_{d(off)}$ is the turn-off delay variation of the corresponding MOSFETs, and $C_{iss}$ stands for the input capacitance. Equation (4) is derived based on the assumption that the gate voltage variation $\delta V_{GS}$ is smaller than 1% of the input gate voltage $V_{GS}$.

$$L_m > \frac{1}{2} \cdot \frac{\delta t^2_{d(off)}}{0.01 \cdot C_{iss}} \tag{4}$$

The magnitude of $\delta V_{GS}$ depends on the amount of discharge $\delta Q_m$ of $C_{iss}$ (undelayed) by the magnetizing current $i_m$ shown in (5). Based on the equivalent inductor circuit model, the $C_{iss}$ of the undelayed switch is discharged by the present magnetizing current $i_m$ and the gate sink current $i_g$. However, the $C_{iss}$ of the delayed switch is discharged only by $i_g$. To synchronize the gate discharge currents, the value of $L_m$ should be sufficiently large to suppress the magnetizing current $i_m$. It should be noted that the total voltage applied on the magnetizing inductance $L_m$ is $V_{GS}$ during the turn-off delay period.

$$\delta V_{GS} = \frac{\delta Q_m}{C_{iss(undelayed)}} = \frac{i_{m(pk)} \delta t_{d(off)}}{2C_{iss(undelayed)}} i_{m(pk)} = \frac{\delta t_{d(off)} V_{GS}}{L_m} \tag{5}$$

Combining the formulas in (5) eventually leads to (4). If the value of $\delta t_{d(off)}$ is relatively small (e.g., 50 ns), the assumption that $\delta V_{GS}$ should be smaller than 1% of $V_{GS}$ leads to a reasonable size of $L_m$. However, if the value of $\delta t_{d(off)}$ is quite large (e.g., 250 ns), the allowable gate voltage mismatch should be increased to avoid having an unrealistically large $L_m$. The magnitude of $\delta V_{GS}$ should then be smaller than 5% of $V_{GS}$. Thus, the denominator of (4) is modified as $0.1 \cdot C_{iss}$. $R_g$ stands for the total gate resistance, including the internal resistance of the gate driver.

$$L_k \leq \frac{C_{iss} R_g^2}{4\xi^2} = \frac{C_{iss} R_g^2}{0.64} \tag{6}$$

The leakage inductance $L_k$ of the gate-coupled inductors can be seen as parasitic inductances in the wires between the gate drivers and MOSFETs. The value of $L_k$ should be designed to prevent the initial and induced gate currents from severe oscillations generated by $L_k$ and $C_{iss}$. The desired tiny $L_k$ can be difficult to realize if the damping factor $\xi$ is chosen to be relatively large ($\geq 0.7$ in [35]). Usually, the integrated gate driver IC has some internal output resistance of around 5 $\Omega$, and an external gate resistance of 10 $\Omega$ is suggested, leading to a total value of $R_g = 15$ $\Omega$. Assuming $\xi$ equals or exceeds 0.4, the required leakage can be realized and the computed $k$ is more than 0.9999.

### 3.1. Experiments with Two Series-Connected MOSFETs Using the GBC Method

In [32], it is shown that for the case of two series-connected power switches, dynamic $V_{DS}$ sharing can be achieved with a turn-off delay variation $\delta t_{d(off)}$ of up to 80 ns using the GBC method. Some experiments are performed to investigate the $V_{DS}$ sharing when a longer $\delta t_{d(off)}$ of up to 560 ns exists, further demonstrating the robustness of the GBC method.

Figure 8 shows the experimental set-up with a two-switch MOSFET string using the GBC method. The switching frequency of this prototype is 2.5 kHz. For the case of two MOSFETs operating in series, compared with the bottom switch, the top one has an additional 560 ns turn-off delay, as shown in Figure 9. The coupled inductor is constructed with interleaved windings, resulting in a coupling factor of $k = 0.9999$. The winding inductances $L_{se}$ are 4.71 mH (white) and 4.56 mH (red), while the leakage inductances $L_k$ are 390 nH (white) and 373 nH (red), shown in the subfigure in Figure 9.

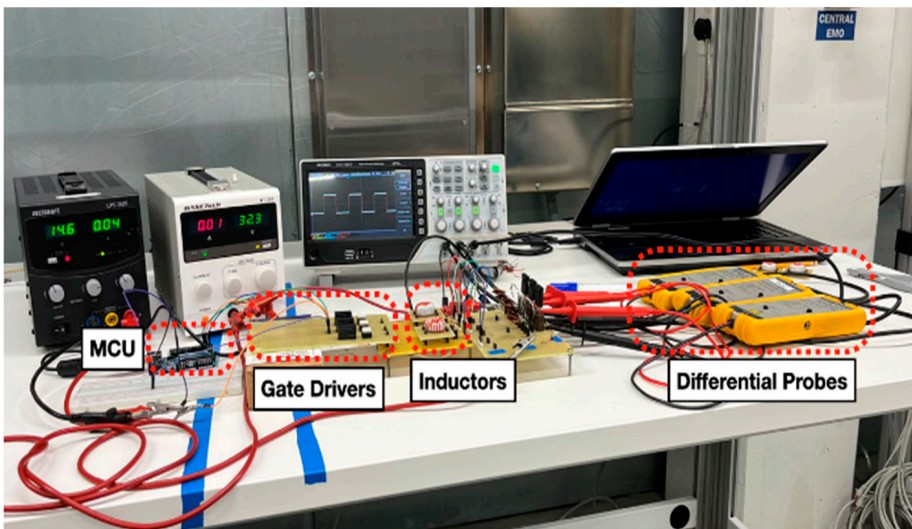

**Figure 8.** Set-up of the series-connected SiC MOSFETs using the GBC method.

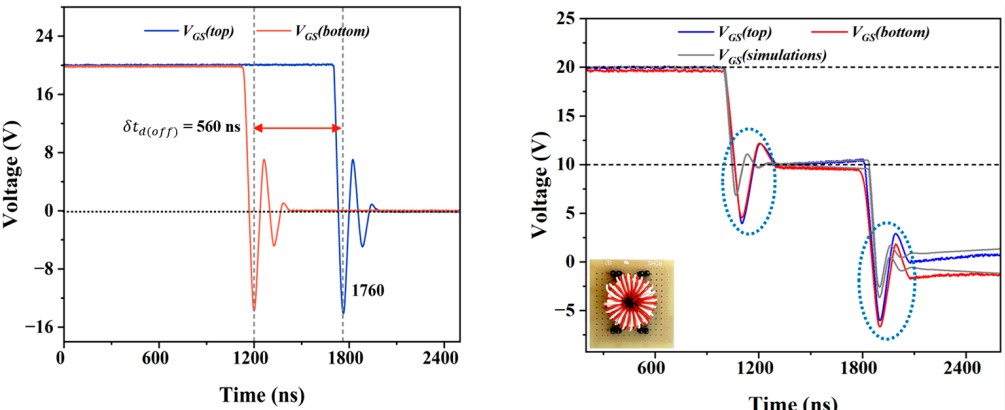

**Figure 9.** Measured $V_{GS}$ waveforms of the two series-connected MOSFETs during the turn-off period without (**left**) and with the gate-coupled inductor with simulation verifications (**right**).

Figure 9 illustrates how the presence of the coupled inductor influences the gate voltages. Before using the coupled inductor, an extra 560 ns delay results in a noticeable difference in turn-off time. After inserting the coupled inductor in the gate circuits, the gate voltages of the MOSFET string are almost perfectly synchronized and balanced at half of the input gate voltage. At the end of the delay $\delta t_{d(off)}$, $V_{GS(top)}$ (delayed) is slightly higher than $V_{GS(bottom)}$ (undelayed). This is explained in Section 3.2 using analytical calculations.

The gate voltages balance at $0.5V_{GS}$ during $\delta t_{d(off)}$ because the magnetizing inductance is sufficiently large to suppress $i_m$. This limits the gate voltage variation to <5% of $V_{GS}$ and $V_{GS(top)} \approx V_{GS(bottom)}$. Since $i_m$ is negligible, if the turns ratio is set to be 1:1, the $C_{iss}$ of the delayed MOSFET and that of the undelayed MOSFET are discharged by identical gate currents $i_g$. Hence, the gate currents of all MOSFETs are synchronized, and the induced voltage $V_{T(top)}$ is equal to the lower winding voltage $V_{T(bottom)}$. According to (7), the sum of $V_{GS(top)}$ and $V_{GS(bottom)}$ is the input gate voltage. Therefore, the final balance point is $0.5V_{GS}$.

$$\text{Delayed}: \ V_{GS} = V_{GS(top)} + V_{T(top)}; \ \text{Undelayed}: \ 0 = V_{GS(bottom)} - V_{T(bottom)}; \quad (7)$$

The parasitic parameters slightly distort the obtained $V_{GS}$ waveforms. However, this issue can be tackled by using two reverse-biased placed diodes in series with the external gate resistors [36] circled in red in Figure 6. Moreover, the comparison in Figure 10

demonstrates that the GBC method can significantly improve the $V_{DS}$ sharing of the SiC MOSFET string in case of the presence of a large $\delta t_{d(off)}$.

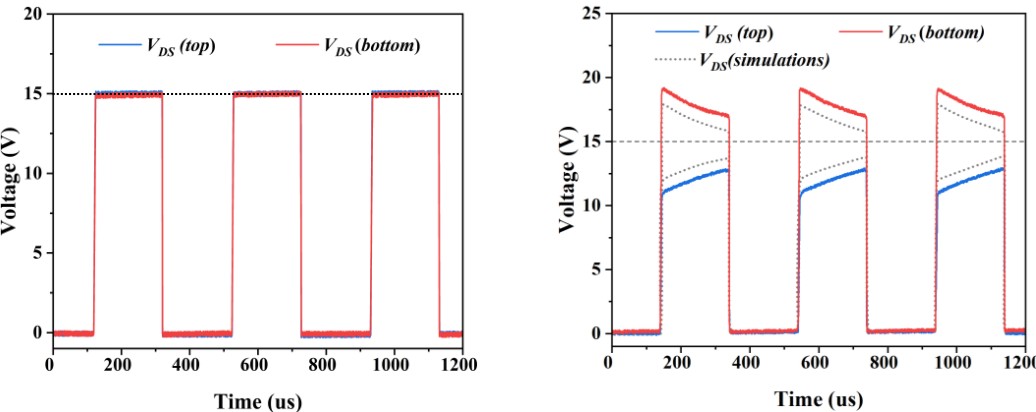

**Figure 10.** Measured $V_{DS}$ of the MOSFET string with (**left**) and without coupled inductor with simulation verifications (**right**).

From a commercial perspective, $N$ series-connected SiC MOSFETs require $(N-1)$ coupled inductors, which can be expensive and result in a bulky solution. However, the circuit and gate-side routing are simple, which reduces the cost of components and manufacturing. Moreover, the required insulation level of the gate-coupled coupled inductor is low ($V_{input}/N$); thus, the cost of each component is reasonable. This means that the GBC method is quite commercially attractive.

### 3.2. Analytical Analysis of Dynamic Voltage Sharing Using the GBC Method

Balanced dynamic voltage sharing of the MOSFET string can be achieved through the GBC method in case of a large $\delta t_{d(off)}$, as shown in Figure 10 (left). To better understand the performance of the MOSFET $V_{GS}$ waveforms during $\delta t_{d(off)}$ with the coupled inductors, an analytical method is derived based on the equivalent gate circuits of the two series-connected SiC MOSFETs during $\delta t_{d(off)}$, as shown in Figure 11.

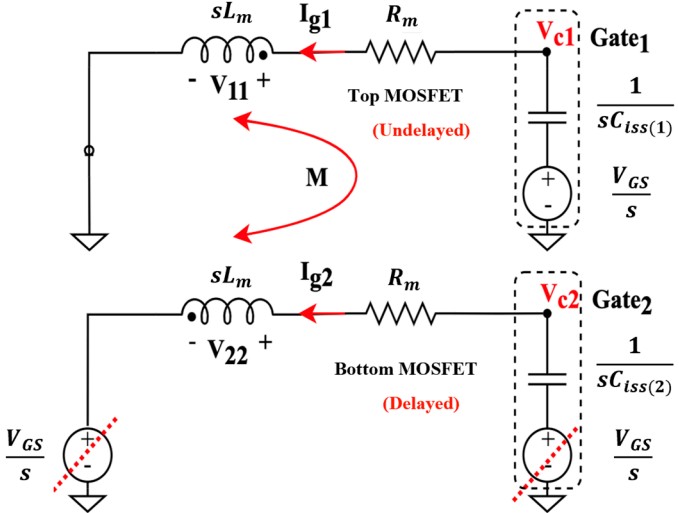

**Figure 11.** Equivalent gate circuits of the two series-connected SiC MOSFETs using GBC method.

Compared with the top MOSFET, the bottom one is assumed to have some extra turn-off delay. Hence, from Figure 11, the bottom MOSFET remains in the on-state. Meanwhile, the top MOSFET is switched off. It can also be noted that the MOSFET gate circuits are simplified as the charged dynamic input capacitors ($C_{iss(1)}$ and $C_{iss(2)}$) circled by black dotted

lines. The analytical method is derived for the case with perfect coupling ($k = 1$, similar to the previous experimental case). Consequently, the self-inductance of the windings will become equal to the magnetizing and mutual inductance ($L_{se} = L_m = M$).

Figure 11 indicates the propagation direction of the gate discharge currents ($I_{g1}$ and $I_{g2}$) and the polarity of the inductor winding voltages. Due to the difference in polarity, the voltages across the upper ($V_{11}$) and lower ($V_{22}$) winding can be calculated from (8).

$$\begin{aligned} V_{11} &= sL_m I_{g1} - sM I_{g2} = sL_m\left(I_{g1} - I_{g2}\right) \\ V_{22} &= sL_m I_{g2} - sM I_{g1} = sL_m\left(I_{g2} - I_{g1}\right) \end{aligned} \tag{8}$$

The input voltage of the bottom MOSFET gate circuit is provided by the gate driver (left), and the remaining voltage on $C_{iss(2)}$ (right) is canceled. Hence, the voltage across the inductor lower winding can be calculated as $V_{22} = -I_{g2}\left(R_m + \frac{1}{sC_{iss}}\right)$. The relationship between the gate sink currents $I_{g1}$ and $I_{g2}$ is shown in (9). Furthermore, the MOSFET gate voltages are calculated using (10).

$$\frac{I_{g2}}{I_{g1}} = \frac{sL_m}{sL_m + R_m + \frac{1}{sC_{iss}}} = \frac{s^2 L_m C_{iss}}{1 + s^2 L_m C_{iss}} \tag{9}$$

$$V_{c(i)} = \frac{V_{GS}}{s} - \frac{I_{g(i)}}{sC_{iss(i)}} \quad (\mathrm{i} = 1,\ 2) \tag{10}$$

If the winding resistance $R_m$ is neglected, the gate discharge current $I_{g1}$ for the gate circuit of the top switch (undelayed) can be derived using (11) and (12).

$$V_{11} + I_{g1} R_m + \frac{I_{g1}}{sC_{iss}} = \frac{V_{GS}}{s} \tag{11}$$

$$I_{g1} = V_{GS} \frac{C_{iss}\left(1 + s^2 L_m C_{iss}\right)}{1 + 2s^2 L_m C_{iss}} \tag{12}$$

According to (9) and (12), the formula of $I_{g2}$ can be derived as shown in (13). After the combination of (10) and (12), and (10) and (13), the gate voltages of the two involved MOSFETs can be derived as (14).

$$I_{g2} = V_{GS} \frac{s^2 L_m C_{iss}^2}{1 + 2s^2 L_m C_{iss}} \tag{13}$$

$$\begin{aligned} V_{c1} &= V_{GS} \frac{sL_m C_{iss}}{1 + 2s^2 L_m C_{iss}} \\ V_{c2} &= V_{GS}\left[\frac{1}{s} - \frac{sL_m C_{iss}}{1 + 2s^2 L_m C_{iss}}\right] \end{aligned} \tag{14}$$

At the beginning of $\delta t_{d(off)}$, the frequency content of the gate voltage waveforms ($V_{GS(top)}$ and $V_{GS(bottom)}$) is large, and the gate current $I_{g1}$ is almost the same as $I_{g2}$ based on (9). Thus, based on (10), the value of $V_{GS(top)}$ should also be the same as $V_{GS(bottom)}$. On the other hand, when the high-frequency harmonics decay, the magnitude of $I_{g2}$ will become smaller than $I_{g1}$, and $V_{GS(bottom)}$ will be larger than $V_{GS(top)}$. These obtained results verify that during $\delta t_{d(off)}$, $V_{GS(bottom)}$ (delayed) first resonates synchronously with $V_{GS(top)}$ (undelayed) but becomes larger than $V_{GS(top)}$ after the oscillations, as shown in Figure 12. As a result, the experimental result shown in Figure 9 (right) matches the analytical results.

If the coupling factor of the inductor is close to 1, the leakage inductance $L_k$ will be minimal, and the corresponding $V_{GS}$ oscillation period will also be small. When the oscillations decay, the gate voltage variation $\delta V_{GS}$ will occur. Conversely, if $k$ is relatively low, the oscillation period $T_o$ will be much larger, and the gate voltages can be precisely synchronized during the entire period of $\delta t_{d(off)}$.

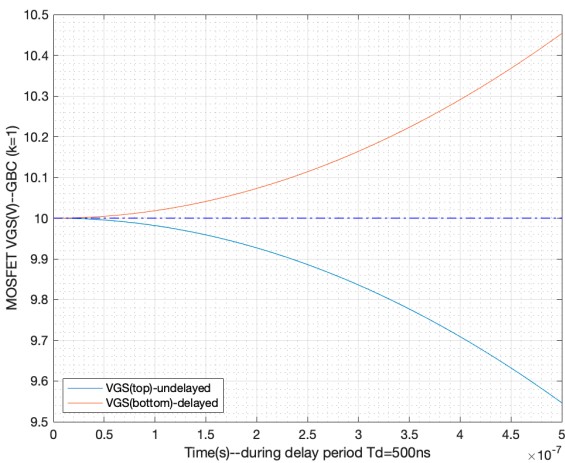

**Figure 12.** $V_{GS}$ curves of the MOSFET string during the turn-off delay with coupled inductor ($k = 1$).

### 3.3. Simulation Verification of Dynamic Voltage Sharing Using the GBC Method

The analytical calculations were verified against the obtained experimental results. The behavior of the $V_{GS}$ waveforms during the entire $\delta t_{d(off)}$ was verified using Simulink using the model shown in Figure 13, which only focuses on the gate circuits of the SiC MOSFET string. The parameters of the simulation model are all extracted from the experiments: the bottom SiC MOSFET has an extra 500 ns turn-off delay compared to the top MOSFET, and the self-inductance of both inductor windings is $L_{se} = 4.56$ mH. The dynamic input capacitance of the MOSFETs is 289 pF (IMW120R220M1H).

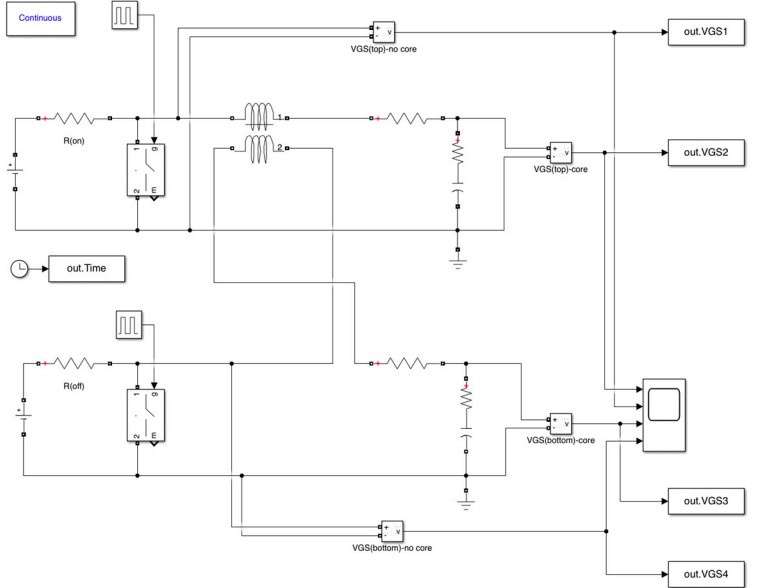

**Figure 13.** The MATLAB Simulink model (gate circuits of two series-connected SiC MOSFETs) built based on the schematic of Figure 11.

If $k$ has a relatively low value, the leakage inductance $L_k$ has a larger value. Therefore, in Figure 14 (left), the resonant frequency of the gate waveforms $f_o$ is lower than in Figure 14 (right). During the period of $\delta t_{d(off)}$, the resonances (generated by $L_k$ and $C_{iss}$) will last; thus, both $V_{GS(top)}$ and $V_{GS(bottom)}$ will be dominated by the HF harmonics and react synchronously. If the value of $k$ is large, almost identical $V_{GS}$ sharing can be observed. The simulated waveforms match quite well with those measured in Figure 9.

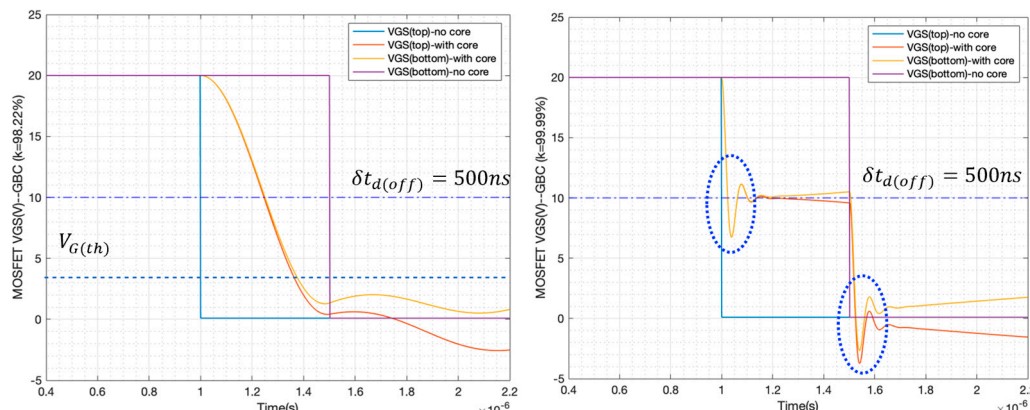

**Figure 14.** $V_{GS}$ waveforms of the two series-connected SiC MOSFETs using GBC method during off-period with inductor $k = 0.9822$ (**left**) and $k = 0.9999$ (**right**) performed in MATLAB Simulink.

The balance point at $0.5V_{GS}$ can only appear if $L_k$ is sufficiently small. If $k$ is poor, the gate pulse resonances will last for the entire duration of $\delta t_{d(off)}$, and thus the $0.5V_{GS}$ balance point is obscured by the oscillations. If $k$ is close to perfect, the resulting $f_o$ is larger. During the period of $\delta t_{d(off)}$ the gate waveforms will first resonate synchronously, but they may deviate for a short period of time after the oscillations have attenuated.

An LTspice simulation model was also built based on the schematic of Figure 6 to verify the $V_{DS}$ waveforms. The component parameters are the same as those of the experiments and Simulink simulations. Again, there is an extra 500 ns turn-off delay in the bottom SiC MOSFET.

The LTspice simulation results show $V_{GS}$ behavior similar to the Simulink model in Figure 13. It should be noted that if the coupling factor of the gate-coupled inductor is low, the $V_{DS}$ sharing of the SiC MOSFET string is poor. Despite the synchronous gate voltages, the $V_{DS}$ rising slew rates of the MOSFETs are different due to the non-identical gate sink currents ($I_{g(delayed)} < I_{g(undelayed)}$), which causes an imbalance in $V_{DS}$, as shown in Figure 15. However, if the value of $k$ is close to 1, the gate sink currents are almost identical. Thus, balanced $V_{DS}$ sharing can be achieved.

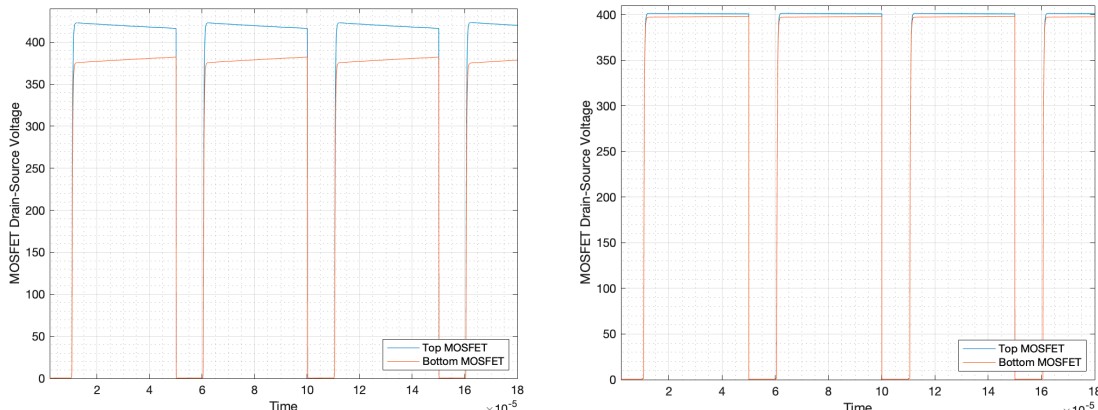

**Figure 15.** $V_{DS}$ waveforms of the SiC MOSFET string using GBC method during off-period with coupled inductor $k = 0.9822$ (**left**) and $k = 0.9999$ (**right**) in case of $\delta t_{d(off)} = 500$ ns (LTspice).

## 4. Hybrid Gate Signal Delay Adjustment Method—Improved RC Snubber Method

The GBC method is solely based on gate signal delay adjustment. Combining GBC with a passive snubber yields the improved RC snubber method. Two different types of improved RC snubber methods were evaluated and validated by experiments and LTspice simulations. Based on the results, the strengths and weaknesses of each method are provided, and a comparison of these two methods is given.

### 4.1. Passive Snubber Circuits

To enhance the $V_{DS}$ sharing of the SiC MOSFET string, passive snubber circuits (e.g., RC and RCD snubbers, as shown in Figure 16) are commonly employed alongside the static balancing resistors. While the RCD snubber may increase cost, it offers significantly lower snubber losses than the RC snubber. During turn-off, the snubber capacitor $C_{sn}$ is charged through the low-impedance diode. During turn-on, it is discharged through the snubber resistor and MOSFET. Thus, the losses induced during turn-off are nearly eliminated. It is important to note that the snubber resistor $R_{sn}$ can provide damping of ringing and voltage spikes on $V_{DS}$ and prevents discharging $C_{sn}$ directly through the MOSFET itself.

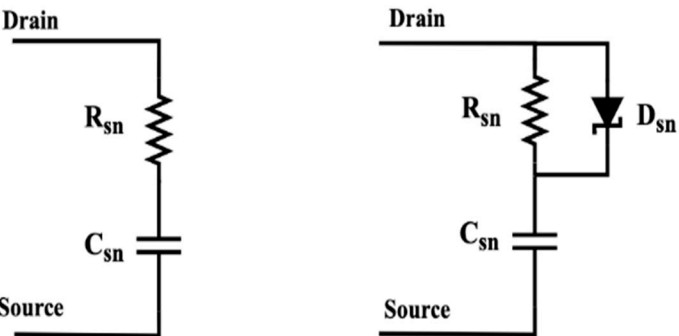

**Figure 16.** RC snubber circuit (**left**) and RCD snubber circuit (**right**).

Usually, the tail-current characteristics, as shown in Figure 17, will exist in IGBTs during the turn-off period. These tail currents impact the voltage sharing of the series-connected IGBTs as the value of the tail-period $T_{tail}$ is unpredictable and deviates from one device to another. Since the SiC MOSFETs do not exhibit a tail-current characteristic, no further discussion related to this factor will be provided.

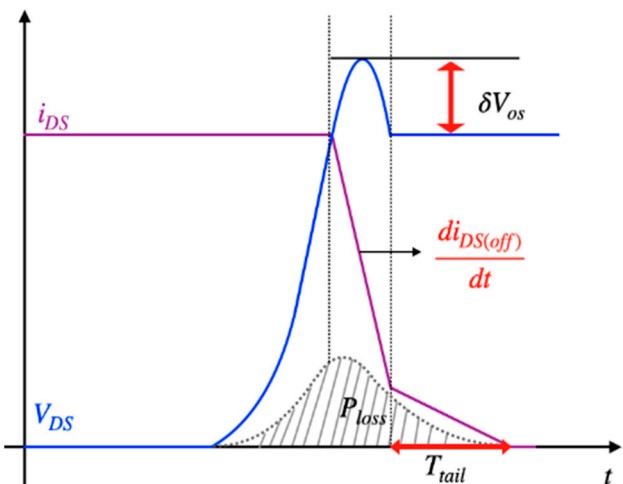

**Figure 17.** $V_{DS}$ and $i_{DS}$ waveform of a IGBT during turn-off period.

If the series-connected SiC MOSFETs run under HF applications, the high drain-current slew rate $di_{DS(off)}/dt$ during turn-off results in a significant voltage overshoot $\delta V_{os}$, as shown in Figure 17. The formula for the magnitude of the overshoot is given in (15), where $L_s$ stands for the total parasitic inductance of the commutation loop.

$$\delta V_{os} = L_s \frac{di_{DS(off)}}{dt} \tag{15}$$

In [37], Baraia introduces the passive clamping snubber shown in Figure 18 (left). In this passive clamping snubber circuit, the snubber capacitor $C_{sn}$ reduces the $V_{DS}$ rising slew rate, and the presence of $R_a$ accelerates the discharging process of $C_{sn}$. During the MOSFET off-period, if $V_{DS}$ exceeds the Zener voltage $V_z$, it gets clamped at $V_z$. This clamping action is crucial in maintaining the drain-source voltage below the device breakdown.

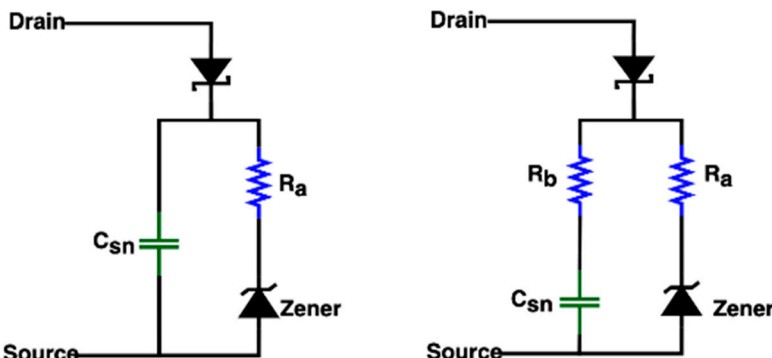

**Figure 18.** Schematic of two types of passive clamping snubber circuit.

Furthermore, in [38], Zhang introduces another resistor, $R_b$, in series with $C_{sn}$, as shown in Figure 18 (right), which assists in reducing the magnitude of the peak voltage across the snubber capacitor. The Zener diode keeps the voltage imbalance $\delta V_{DS}$ within the limits, making the entire circuit more reliable and robust.

The four types of passive snubber circuits discussed and evaluated in the following paragraphs all slow down the $V_{DS}$ slew rate of the series-connected SiC MOSFETs to reduce the $\delta V_{DS}$ generated by $\delta t_{d(off)}$. Usually, better suppression of $\delta V_{DS}$ can be reached for larger snubber capacitance values. However, to ensure sufficient switching speed and lower switching losses of the SiC MOSFETs, the gate signal delay adjustment method should be combined with the passive snubber circuit.

*4.2. Improved RC Snubber Method (a)*

In [33,39], the improved RC snubber method (a) was described. The critical part of this technique is to use a coupled inductor whose primary windings are coupled with the passive snubber circuits to generate a compensation voltage on the secondary winding, which behaves as feedback to the gate, as shown in Figure 19. Through the change in gate voltage, the MOSFET $V_{DS}$ rise slew rate can be altered, and balanced voltage sharing of the SiC MOSFET string can be achieved.

In Figure 19, two coupled inductors are connected to the RC snubber and gate circuits of the SiC MOSFET string. Note that the polarities of the two secondary windings are different. If the bottom SiC MOSFET has an extra 200 ns turn-off delay, the top one will turn off faster, and some capacitive currents will flow through the top snubber circuit. Feedback voltages will be induced on both secondary windings, which are added to the original gate voltages. The feedback voltage of the top SiC MOSFET enhances its overall gate voltage but decreases the slew rate of $V_{DS}$. In contrast, the feedback voltage of the bottom MOSFET decreases its overall gate voltage but increases the slew rate of $V_{DS}$.

An LTspice simulation model was built based on the schematic of Figure 19 to verify the voltage-sharing performance. Both coupled inductors have a primary inductance $L_p$ of 880 µH and a secondary inductance $L_s$ of 24 µH with a high coupling factor. The external gate resistance $R_{g(i)}$ is 25 Ω, and the snubber capacitance $C_{sn}$ and resistance $R_{sn}$ are selected as 330 pF and 50 Ω, respectively.

The passive RC snubber circuit improves the $V_{DS}$ sharing of the two series-connected SiC MOSFETs in case of a large turn-off delay variation. Nevertheless, a considerable voltage imbalance $\delta V_{DS}$ remains. Comparing Figure 20 (left) to Figure 21 (left), the improved RC snubber (a) shows a significantly improved $V_{DS}$ sharing capability. With the adoption

of RC snubber method (a) in Figure 21, good $V_{DS}$ sharing can be achieved with a maximum imbalance voltage of just 2% of $V_{DS}$.

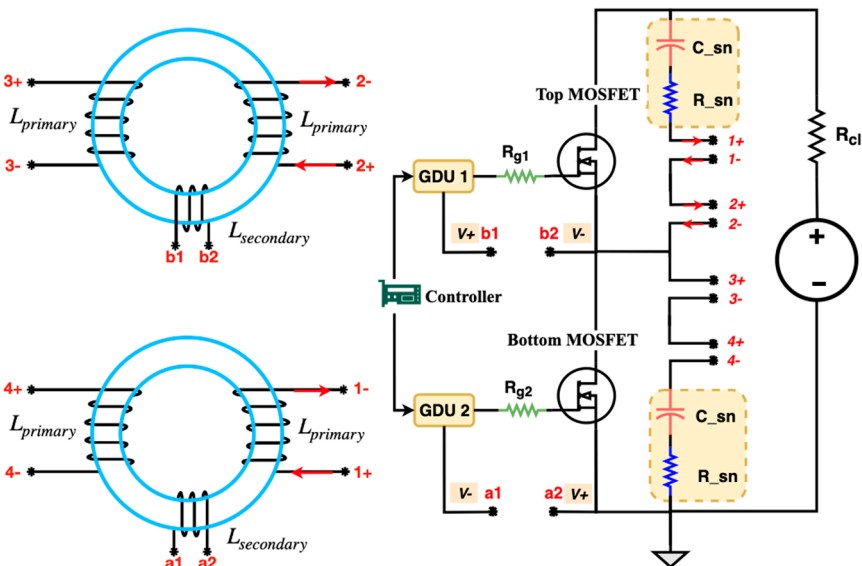

**Figure 19.** Schematic of two series-connected MOSFETs using improved RC snubber method (a).

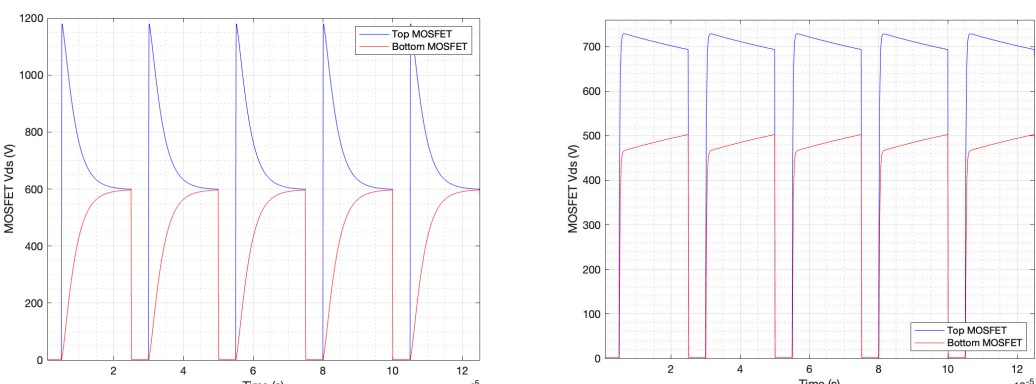

**Figure 20.** $V_{DS}$ sharing of the SiC MOSFET string using a passive RC snubber with an extra 200 ns turn-off delay in the bottom switch without snubber (**left**) and with snubber (**right**).

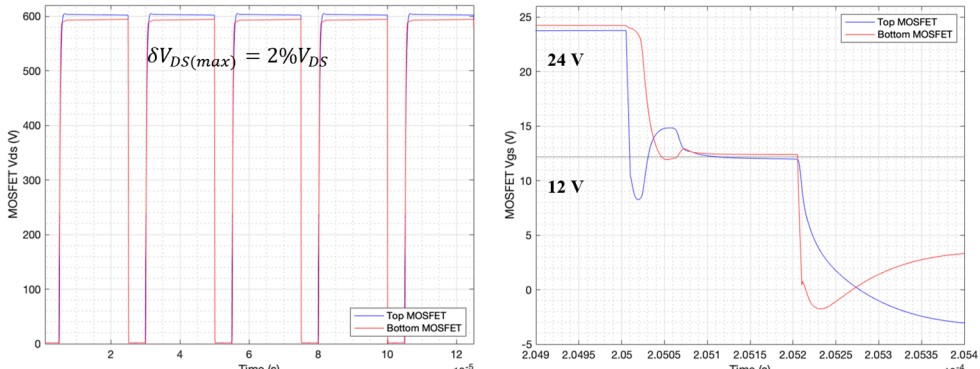

**Figure 21.** $V_{DS}$ (**left**) and $V_{GS}$ during turn-off period (**right**) sharing of the SiC MOSFET string using the improved RC snubber method (a) with an extra 200 ns turn-off delay in the bottom switch with inductor.

Figure 21 (right) demonstrates the $V_{GS}$ waveforms of the SiC MOSFET string during the turn-off transition. The variation in gate-source voltage $\delta V_{GS}$ is almost eliminated using the coupled inductors. Despite slight $V_{GS}$ waveform distortion due to parasitic oscillations, the gate voltages switch almost synchronously and are balanced at $0.5V_{GS}$. If the feedback voltages induced on the secondaries are sufficient, the gate currents can be synchronized. Thus, the gate charge movement velocity of the MOSFET from the string is almost identical and $\delta V_{GS}$ is suppressed.

In conclusion, the improved RC snubber method (a) results in good $V_{DS}$ sharing of the SiC MOSFET string. The disadvantage, however, is the relatively large number of required components, which leads to a higher cost ($N$ series-connected MOSFETs require $N$ coupled inductors) and complicated snubber circuit routing. Finally, the scalability of this method (a) is low, which limits the usable input voltage.

### 4.3. Improved RC Snubber Method (b)

Figure 22 shows the schematic of the improved RC snubber method (b), an optimized variant of method (a), applied to two series-connected MOSFET. A four-port coupled inductor links two snubber circuits and two gate circuits. The main difference with method (a) is that only one primary winding is coupled within one snubber branch. Hence, the circuit layout and routing of method (b) are much more straightforward. Moreover, one fewer inductor is required, reducing the total material cost. The polarity of the inductor secondary windings still needs to be different to generate the gate feedback voltages with opposing polarity.

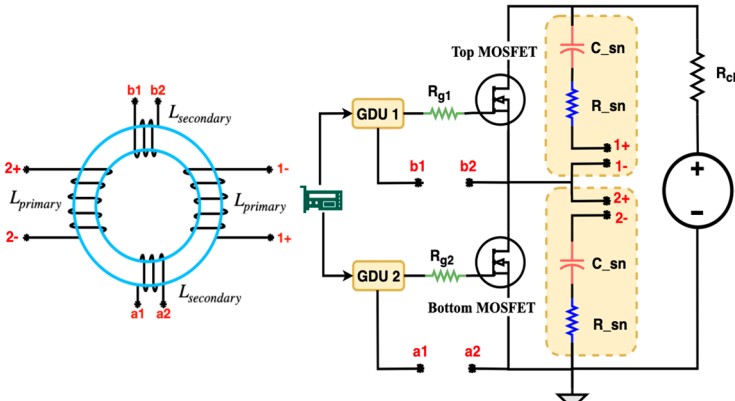

**Figure 22.** Schematic of the two series-connected MOSFETs using improved RC snubber method (b).

Comparing the simulation results in Figure 23 to Figure 24 (left), in case of an additional 200 ns turn-off delay existing on the bottom switch, with the utilization of the four-port inductor, the $V_{DS}$ sharing of the SiC MOSFET string is again excellent. The maximum voltage imbalance is just 1.5% of $V_{DS}$, which is slightly better than that obtained using method (a). This degree of voltage imbalance is negligible and cannot lead to device breakdown.

Figure 24 (right) shows the $V_{DS}$ sharing of the SiC MOSFET string during the turn-off delay. After the turn-off of the top MOSFET during the delay period $\delta t_{d(off)}$, the slew rate of $V_{DS(top)}$ (blue) is quite large at the very beginning and then decreases due to the gate feedback voltage $V_{comp(1)}$. Meanwhile, $V_{DS(bottom)}$ (red) remains turned on for around 15 ns and then increases considerably due to the presence of $V_{comp(2)}$. After the delay period, at $t_1$, the bottom switch will be fully turned off, and the slew rate of $V_{DS}$ will grow significantly. The actual turn-off delay period of the bottom switch has been shortened drastically from 200 ns to 15 ns due to the gate compensation circuits (coupled inductor secondaries).

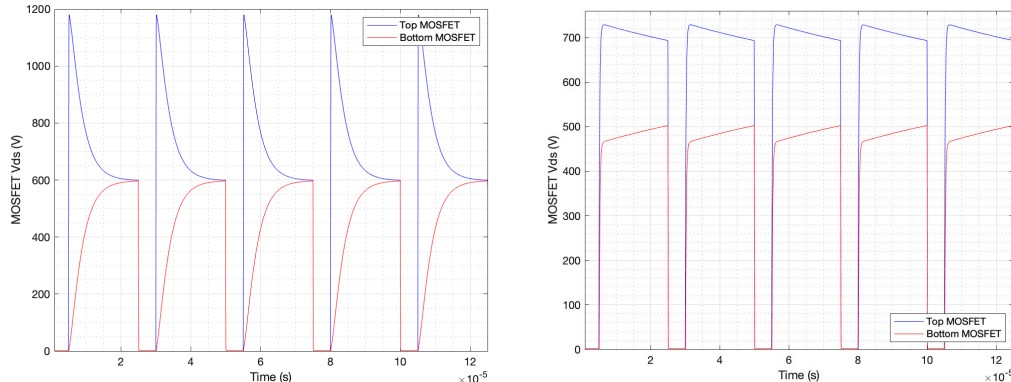

**Figure 23.** $V_{DS}$ sharing of the SiC MOSFET string using the improved RC snubber method (b) with an extra 200 ns turn-off delay in the bottom switch without snubber (**left**) and with snubber without inductor (**right**). Repeated from Figure 20 for better comparison with Figure 24.

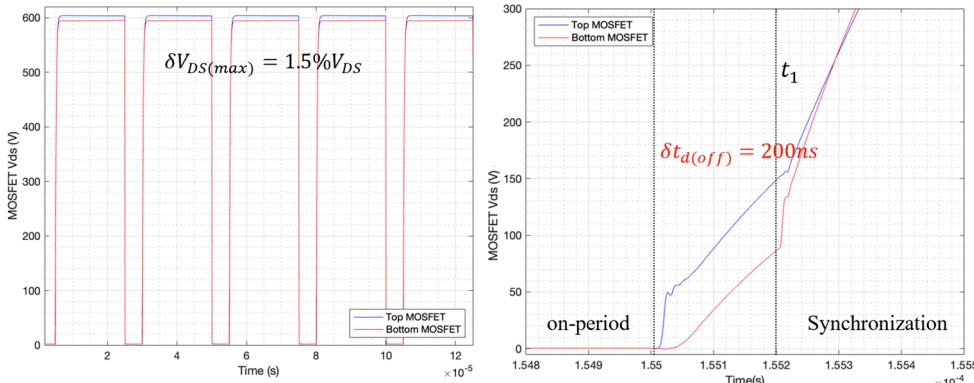

**Figure 24.** $V_{DS}$ (**left**) and $V_{DS}$ during off-period (**right**) sharing of the SiC MOSFET string using improved RC snubber method (b) with a 200 ns turn-off delay in the bottom switch with inductor.

To conclude, both improved RC snubber methods can achieve near-perfect voltage balancing. The maximum imbalance voltage $\delta V_{DS}$ is well within the safety margin and cannot cause the devices to break down. In [33,39], the feasibility of the proposed improved RC snubber method (a) has already been verified by experiments, although the applied MOSFET switching delay variation is only 5 ns. Hence, to obtain convincing evidence that the improved RC snubber method (b) also works well, it is validated through experiments with a larger turn-off delay variation.

### 4.4. Experiments Using Improved RC Snubber Method (b) on Four Series-Connected SiC MOSFETs

The schematic of four series-connected SiC MOSFETs using improved RC snubber method (b) is shown in Figure 25. The top and bottom two series-connected MOSFETs can each be regarded as one equivalent switch. Then, these two equivalent switches are connected in series. Apart from the four-port coupled inductors, a two-port inductor is also coupled within the RC snubber circuits of those two equivalent switches.

Based on the schematic of Figure 25, an LTspice simulation model was built for four series-connected SiC MOSFETs using the improved RC snubber method (b). The coupling factors of the inductors are set to 0.9999. The primary inductance associated with the snubber circuit is 800 µH, and the secondary inductance associated with the gate circuit is 24 µH. The capacitance of $C_{sn(1)}$ and $C_{sn(2)}$ is 330 pF, and that of $C_{sn(3)}$ is 200 pF to guarantee a sufficient voltage slew rate. MOSFETs 3 and 4 have an extra 125 ns turn-off delay, and MOSFET 1 has an additional 250 ns turn-off delay compared to MOSFET 2. The external gate resistance is 25 Ω, and the snubber resistance of the snubber circuits is 50 Ω.

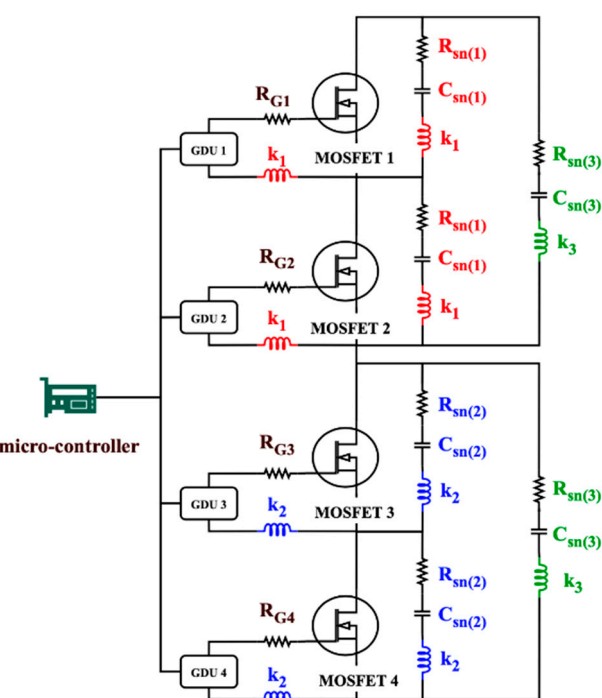

**Figure 25.** Schematic of four series-connected MOSFETs using improved RC snubber method (b).

Figure 26 indicates the $V_{DS}$ sharing of the four series-connected SiC MOSFETs before and after the connection of the three inductors. Comparing Figure 26 (right) to Figure 26 (left) shows that nearly perfect drain-source voltage sharing can be achieved by means of these three coupled inductors. Therefore, the improved RC snubber method (b) is verified to have significantly improved the voltage-sharing capability, even for more than two series-connected MOSFETs.

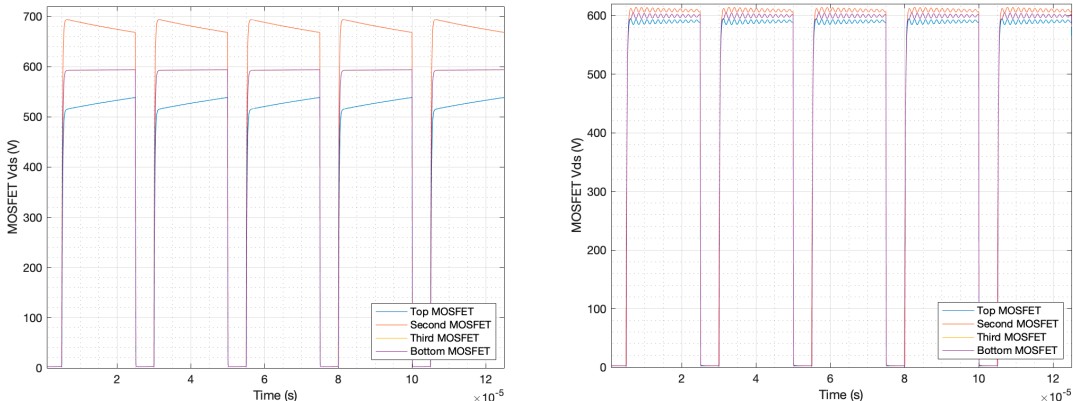

**Figure 26.** $V_{DS}$ sharing of the SiC MOSFET string using the improved RC snubber method (b) with snubber but without inductors (**left**) and with snubber and inductor (**right**).

Figure 27 (left) illustrates the experimental implementation of the same circuit. The gate triggering pulses are shown in Figure 27 (right): MOSFETs 3 and 4 both have two micro-controller clock cycles' turn-off delay variation (125 ns), and MOSFET 1 has four extra micro-controller clock cycles' turn-off delay (250 ns) compared to MOSFET 2. An extra 20 ns switching delay variation is found between the $V_{GS}$ waveforms of MOSFETs 3 and 4. This delay difference is generated by the parameter variation in the gate drivers, the difference in the connection wire length, and the parasitic parameters. Thus, the MOSFETs have different switching delays.

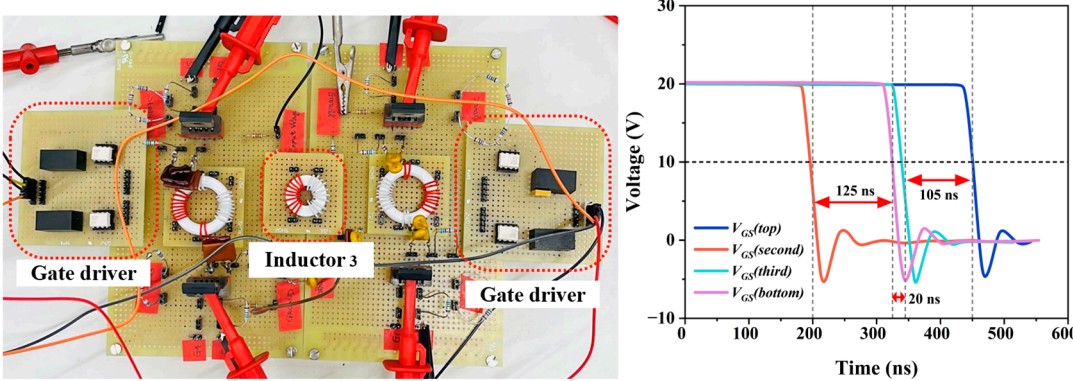

**Figure 27.** Experimental set-up of the four series-connected SiC MOSFETs (**left**) using improved RC snubber method (b) and measured $V_{GS}$ waveforms during off-period (**right**).

Comparing Figure 28 (left) to (right), it can be concluded that the coupled inductors can help to significantly improve the $V_{DS}$ sharing of the four series-connected SiC MOSFETs, even though there is turn-off delay variation. From Figure 28 (right), the $V_{DS}$ sharing among the MOSFET string is nearly perfect, proving the functionality of the improved RC snubber method (b) and validating the feasibility of the circuit in Figure 25.

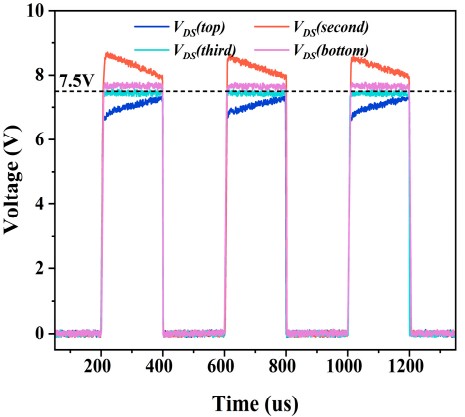 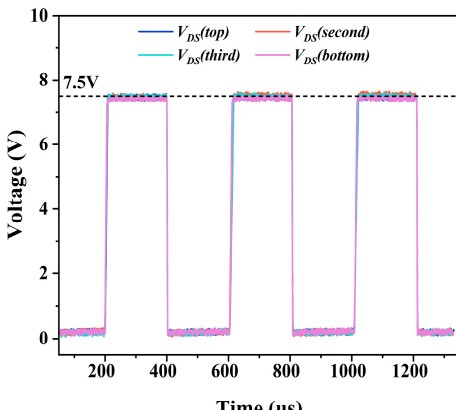

**Figure 28.** Measured $V_{DS}$ sharing of the four series-connected SiC MOSFETs using the improved RC snubber (b) without coupled inductors (**left**) and with coupled inductors (**right**).

As a result, based on Figure 26 (right) and Figure 28 (right), the balanced voltage sharing among the series-connected SiC MOSFETs can be implemented by means of the improved RC snubber method (b). However, the biggest challenge is the insulation design of the coupled inductors, as shown in Figure 25. Inductor 3 must withstand the whole input voltage $V_{in}$, and thus is more sensitive to breakdown. Another drawback of this method is that the number of series-connected MOSFETs cannot be arbitrary but should be a power of two ($N = 2^x$). This imposes strict limitations on the usable voltage.

## 5. Gate-Drain Zener Clamping Circuits

The Zener clamping method is one of the most effective solutions for the unbalanced $V_{DS}$ sharing of the series-connected SiC MOSFETs. Generally, the Zener clamping circuit comprises Zener diodes and some passive components, which are applied between the gate-drain terminals to limit the overvoltage across the SiC MOSFET. The Zener clamping method only addresses the overvoltages caused by voltage imbalance, thereby protecting the semiconductor devices. For optimal results, it is recommended to use this method in conjunction with passive snubber circuits to achieve superior static and dynamic voltage sharing.

The basic Zener clamping method was first proposed in [40], and requires a Zener diode to clamp the $V_{DG}$ of the corresponding MOSFET as a means of overvoltage control. An optimized Zener clamping method was suggested in [41] to achieve satisfactory operation with a high input voltage for the MOSFET string. Multiple diodes are connected in series to increase the clamping voltage of every Zener branch. Multiple series-connected diodes are used because the commercially available Zener diodes are limited to about 400 V. It can be noted that a 400 V Zener diode has a maximum continuous current of 7 mA with a power loss of approximately 0.25 W.

In the basic Zener clamping method, as depicted in Figure 29 (left), the Zener diodes $Z_1$ and $Z_2$ clamp the drain-gate voltages of their respective MOSFETs to the Zener voltage. If the bottom MOSFET experiences some extra turn-off delay, $V_{DG(top)}$ will rise faster than $V_{DG(bottom)}$ while the top MOSFET is switching off. When the value of $V_{DG(top)}$ exceeds the Zener voltage, it is clamped to $V_{Z1}$, thus preventing the breakdown of the top MOSFET.

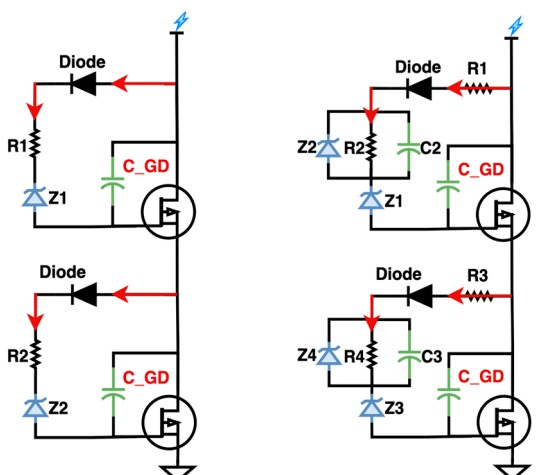

**Figure 29.** Schematic of the SiC MOSFET string using the basic (**left**) and optimized (**right**) Zener clamping circuits.

In practice, the SiC MOSFET reverse capacitance $C_{rss}$ decreases drastically with increasing $V_{DS}$ [42,43]. Since the bottom SiC MOSFET has some extra turn-off delay, after the delay period $\delta t_{d(off)}$, $V_{DG(top)}$ is larger than $V_{DG(bottom)}$, but $C_{gd(top)}$ is smaller than $C_{gd(bottom)}$. Hence, $dV_{DG(top)}/dt$ gradually increases, and the variation between $V_{DG(bottom)}$ and $V_{DG(top)}$ becomes more noticeable.

In [44,45], an optimized Zener clamping method was proposed; see Figure 29 (right). Suppose the bottom MOSFET still has some extra turn-off delay $\delta t_{d(off)}$ during the first commutation period, as shown in Figure 30. In that case, the $dV_{DG(top)}/dt$ will grow rapidly before reaching the first-step Zener clamping voltage $V_{Z1}$ because $C_{rss(top)}$ reduces when $V_{DG(top)}$ increases. If the value of $V_{DG(top)}$ is larger than $V_{Z1}$, during the second commutation period, the capacitor $C_2$ plays a crucial role in slowing down the $dV_{DG(top)}/dt$ to 'wait' for the growth of $V_{DG(bottom)}$. Thus, the variation between $V_{DG(top)}$ and $V_{DG(bottom)}$ will decrease. The optimized Zener clamping method allows the beginning of the commutation to be as quick as possible and slow down at the end. Furthermore, using capacitors $C_2$ and $C_3$ can minimize the effect of varying $C_{gd}$ with voltage and between MOSFET devices. During the second commutation period, the total capacitor $C_{tot}$ used for slowing down the $dV_{DS}/dt$ is $(C_2 + C_{gd(top)})$ or $(C_3 + C_{gd(bottom)})$.

Proper component selection is critical when using the optimized Zener clamping method. Referring to Figure 29 (right), if the expected applied voltage of each SiC MOSFET under balanced $V_{DS}$ sharing is $V_a$, which is 60% of the maximum blocking voltage $(V_{DG} \approx V_a = V_{DS})$, the voltage of $Z_1$ should be slightly smaller but close to $0.5V_a$. Additionally, the sum of $V_{Z_1}$ and $V_{Z_2}$ is the total Zener voltage $V_{Z(tot)}$, which should be slightly smaller but close to $V_a$. These selection criteria are such because some voltage will be

dropped over the resistor $R_1$ and $R_3$, connected in series with the Zener diodes. Generally, the value of $V_{Z(tot)}$ is chosen as $V_a$.

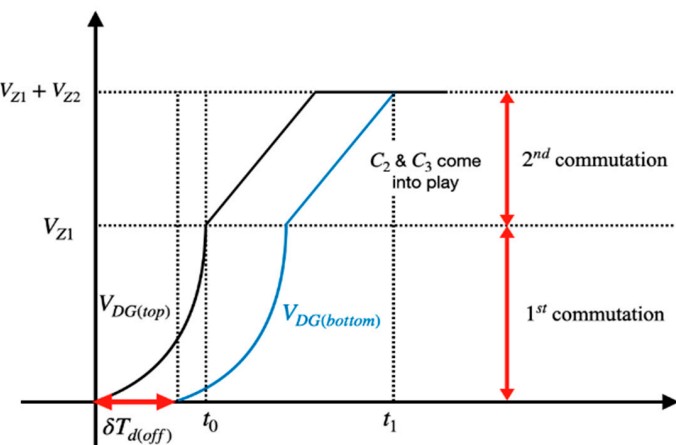

**Figure 30.** Operation principle of two series-connected MOSFETs for optimized Zener clamping method.

During the static balancing period (after $t_1$), the resistors $R_2$ and $R_4$ shown in Figure 29 (right) are dominant and limit the current through the Zener branch. The current $I_r$ that flows through $R_2$ or $R_4$ should be at least 10 times the drain leakage current $I_{DSS}$ found on the datasheet of the selected SiC MOSFET [46].

$$R_2 = \frac{V_a - V_{Z_1}}{I_r} = \frac{V_a - V_{Z_1}}{10 I_{DSS}} \tag{16}$$

$$C_2 \leq \frac{T_{on}}{(3 \sim 5) R_2} \tag{17}$$

During the second commutation period (from $t_0$ to $t_1$), the capacitors $C_2$ and $C_3$ become dominant and slow down the $dV_{DG(top)}/dt$ and $dV_{DG(bottom)}/dt$ if $V_{DG}$ is larger than $V_{Z_1}$. When the values of $R_2$ and $R_4$ are determined based on (16), the capacitance of $C_2$ and $C_3$ can be calculated from (17). The time constant of the branch is chosen to be less than 35 times the on-period $T_{on}$.

*Experiments with the Optimized Zener Clamping Method and Three Series-Connected MOSFETs*

Figure 31 illustrates the experimental set-up used to verify the optimized Zener clamping method. The top SiC MOSFET has an extra 560 ns (8 micro-controller clock cycles) turn-off delay compared to the other MOSFETs, and a switching frequency of 2.5 kHz.

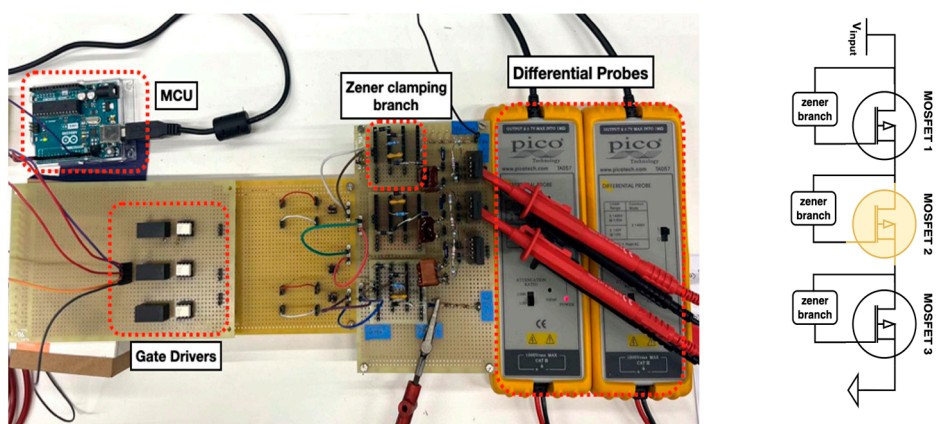

**Figure 31.** Set-up of the three series-connected MOSFETs using Zener clamping method.

As seen in Figure 32 (left), balanced static $V_{DS}$ sharing can be achieved with Zener clamping in case of a large $\delta t_{d(off)}$. However, as shown in Figure 32 (right), completely balanced dynamic $V_{DS}$ sharing has yet to be achieved. The performance of the $V_{DS}$ wavefronts matches the working principle of the optimized Zener clamping method: the $V_{DS}$ first grows rapidly with a high slew rate when it is lower than the first Zener voltage $V_Z$. After this, the slew rate decreases considerably if the value of $V_{DS}$ is larger than $V_Z$ due to the capacitors in the Zener branches. If the value of $V_{DG}$ climbs above $V_{Z(tot)}$, it will be clamped, preventing the destruction of the MOSFET due to overvoltage.

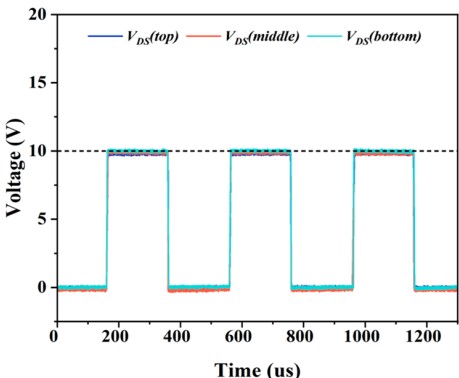 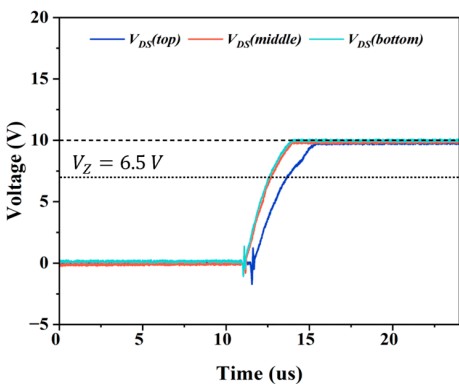

**Figure 32.** $V_{DS}$ overall waveforms (left) and $V_{DS}$ waveforms during off-period of the SiC MOSFET string with optimized Zener clamping circuits ($\delta t_{d(off)} = 560$ ns).

Among the SiC MOSFET string, if there is no $\delta t_{d(on)}$ or $\delta t_{d(off)}$, but some parasitic parameter variation $\delta C_{iss}$ exists, balanced $V_{DS}$ sharing can still be achieved. Figure 33 (left) illustrates this in the case of $\delta C_{iss(2)} = 85$ pF, where the middle SiC MOSFET is replaced by another type, C3M0280090D, with a smaller $C_{iss}$ compared to the selected switch type IMW120R220M1H. From Figure 33 (right), it should be noted that during the MOSFET off-period, the slew rates of $V_{DS(top)}$, $V_{DS(middle)}$, and $V_{DS(bottom)}$ are almost identical, guaranteeing balanced voltage sharing. Thus, the small amount of $\delta C_{iss}$ in the MOSFET string is not considered the leading cause of voltage-sharing imbalance.

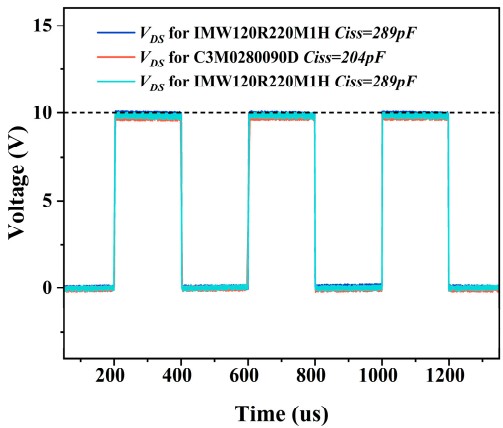 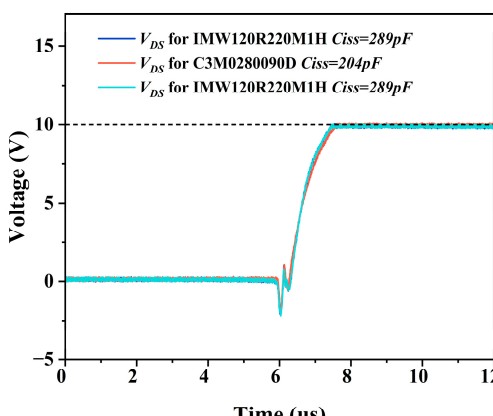

**Figure 33.** $V_{DS}$ overall waveforms (**left**) and $V_{DS}$ waveforms during off-period (**right**) of the SiC MOSFET string with optimized Zener clamping circuits ($\delta C_{iss(2)} = 85$ pF).

In summary, the optimized Zener clamping method can successfully clamp the overvoltage caused by $\delta t_{d(off)}$ to avoid SiC MOSFET breakdown. However, this method cannot improve the dynamic voltage sharing of the SiC MOSFET string. This method only requires some cheap components, which results in a low material cost. As the voltage level increases, more losses will be induced in the Zener branch, consisting of some static dissipation during the off-time, as well as (larger) dynamic losses associated with the clamping current.

## 6. Design of Isolated HV Gate Drivers

While the presented methods can indeed achieve statically and dynamically balanced voltage sharing, another big challenge must be addressed. When *N* SiC MOSFETs are connected in series, as shown in Figure 34, the required gate driving potential for each consecutive switch increases. The gate terminal voltage potential is given in (18). For each MOSFET, the potential of the gate ($V_{G(i)}$) and source ($V_{S(i)}$) terminals is almost the same. From (18), it is apparent that the gate potential of the MOSFETs close to the input lead is relatively high, comparable to the HV input voltage $V_{input}$.

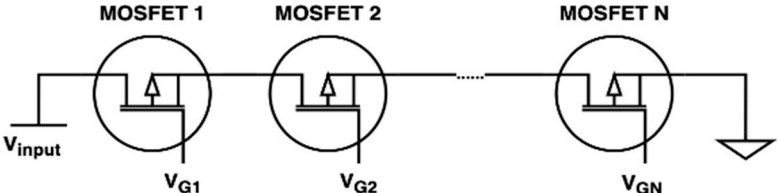

**Figure 34.** Schematic of multiple (*N*) series-connected SiC MOSFETs.

$$V_{G(i)} = \frac{(N-i)}{N} V_{input} \quad (i = 1, \ldots, N) \tag{18}$$

In this scenario, to avoid the breakdown of the gate drivers, gate drivers with a high isolation voltage are required. HV gate drivers (e.g., 8–10 kV level) are not yet commercialized, and the price of the available ones is excessively high. Therefore, the design of such a HV gate driver is essential to achieve a cost-effective HV switch with series-connected MOSFETs.

The proposed solution for enhancing the isolation voltage of the driving circuit is to replace the isolated gate driver with a non-isolated driver, shifting the isolation barrier to an optocoupler and isolated DC/DC converter. Commercial optocouplers are available with an isolation voltage that is much higher than that of isolated gate drivers (beyond 20 kV). In addition, the propagation delay variation of non-isolated drivers is typically better. This concept is illustrated in Figure 35, where the non-isolated gate drivers are implemented using a BJT push–pull stage with split output.

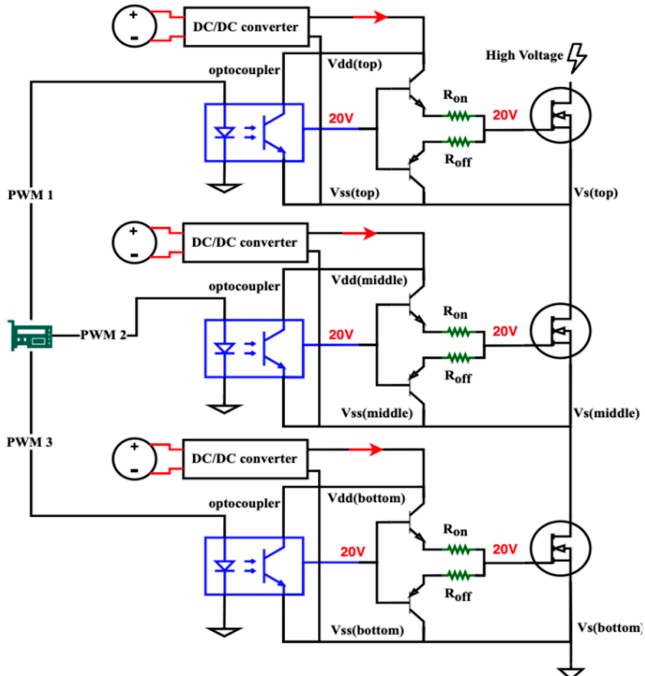

**Figure 35.** Schematic of the SiC MOSFET string driven by the optically isolated HV gate drivers.

A prototype (Figure 36) was built with three series-connected MOSFETs and optically isolated HV gate drivers according to the schematic in Figure 35, using an IXDD630MXI non-isolated driver, OPI1268S optocoupler, and RHV2-0512D isolated DC/DC converter. While the DC/DC converter has an isolation level of 20 kV for 1 s, destructive tests have shown that its maximum continuous working voltage is about 8 kV, thus limiting the input voltage of the MOSFET string.

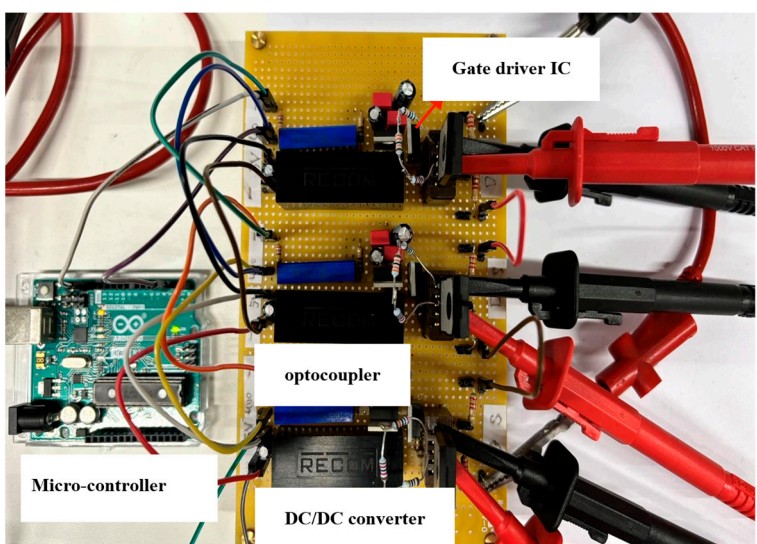

**Figure 36.** The prototype of three series-connected MOSFETs controlled by HV gate driving circuit.

As shown in Figure 37 (left), the static and dynamic voltage sharing is relatively good during the SiC MOSFET off-period, but some imbalance remains. This is due to delay variation in the input PWM signal propagation (e.g., wire length differences) and the variation in gate circuit parameters and component characteristics. Therefore, it is still recommended to use one of the previously described dynamic balancing techniques. Alternatively, the gate circuit components could be binned and matched during production, allowing for operation without a dynamic balancing technique.

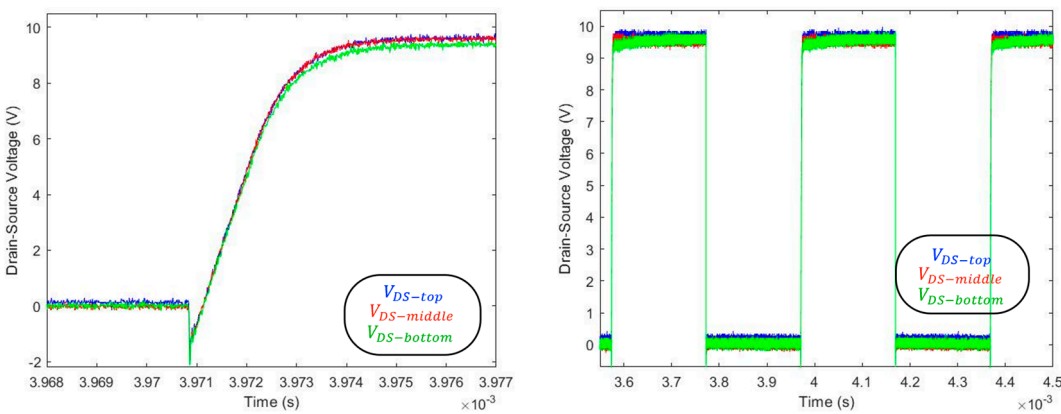

**Figure 37.** Measured $V_{DS}$ waveforms during off-period (**left**) and overall $V_{DS}$ waveforms (**right**).

The results in Figure 37 were obtained from low-voltage tests. Tests were also performed with an input voltage of 2.8 kV (no current limiting resistors), and satisfactory output performance was obtained, as shown in Figure 38. The output voltage was measured using an HV probe. Performing drain voltage measurement on every MOSFET during HV operation was not possible because the commercial differential probes were limited to an isolation level of 1.4 kV.

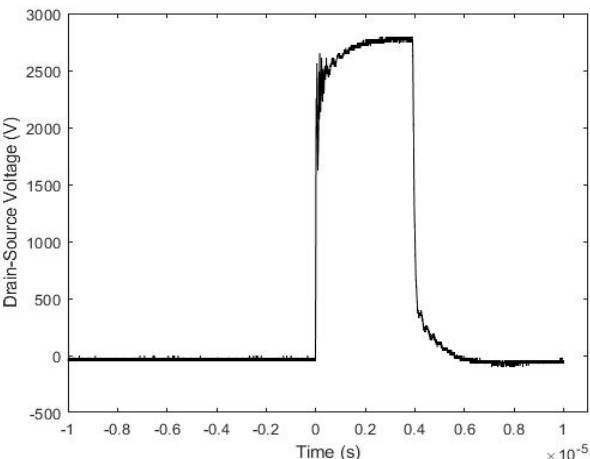

**Figure 38.** Measured $V_{DS}$ waveform of the entire SiC MOSFET string with a voltage of 2.8 kV.

## 7. A Magnetically Isolated HV Gate Driver

The optically isolated gate driver concept is quite simple and provides robust control of the SiC MOSFET string. However, if the number of MOSFETs is large and the input voltage is high, the optocouplers and DC/DC converters must have a high insulation level, which can result in a bulky and expensive construction. A magnetically isolated gate driver is proposed to address these issues [47]. Similar approaches have resulted in HV switches with blocking voltages of 5 to 15 kV [48,49].

Figure 39 (left) presents the schematic of the magnetically isolated driver. The key element of this method is the use of two gate-coupled transformers. One transformer is for turning on, while the other is for turning off the HV MOSFET. Each transformer features an HV-insulated wire as the primary winding, with the secondary winding wound on the toroidal cores as shown in Figure 39 (right).

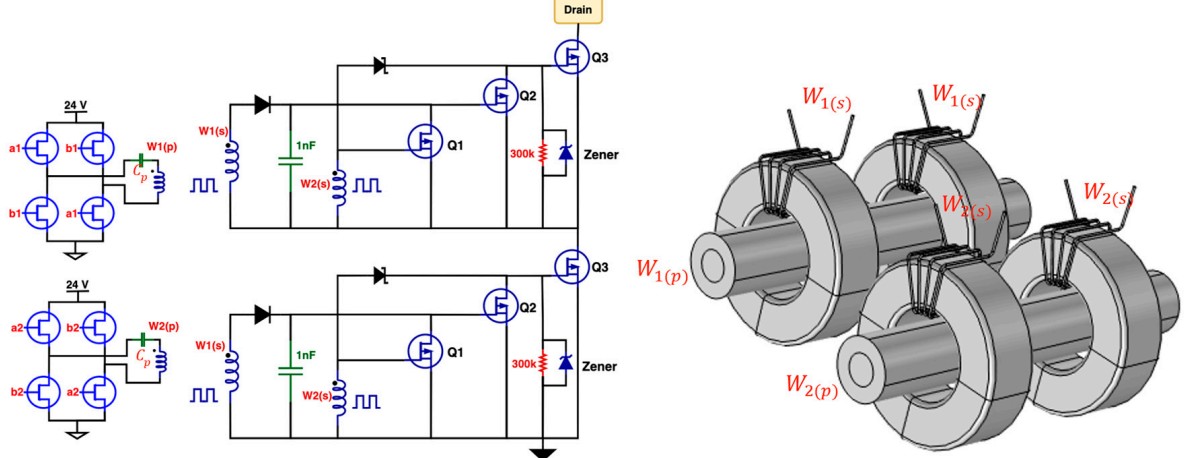

**Figure 39.** Schematic of the magnetically isolated gate driver based two series-connected MOSFETs (**left**) and two gate-coupled transformers (**right**).

The construction shown in Figure 39 allows for the synchronous delivery of the gate pulses to the corresponding auxiliary switches ($Q_1$ or $Q_2$). Hence, the contribution of turn-on and turn-off delay variation $\delta t_d$ caused by propagation path differences is eliminated. As with the optically isolated gate driver, the isolation barrier is moved from the driver circuitry to the transformers. If an HV-insulated wire is used for the primary winding, it can handle large input voltages applied to the HV switch.

To avoid saturation of the toroidal cores, the pulses applied to the transformer primary wires $W_{1(p)}$ and $W_{2(p)}$ should be bipolar. Moreover, for the proper operation of the HV MOSFETs ($Q_3$), the transformer input pulses delivered to their corresponding auxiliary switches $Q_1$ and $Q_2$ should be complementary. That is, the turn-on and turn-off signals are modulated using on–off keying, as shown in Figure 40. Because the waveform is bipolar with a frequency that is independent of the desired on–off timing, there is no risk of core saturation and arbitrarily long on-times can be achieved.

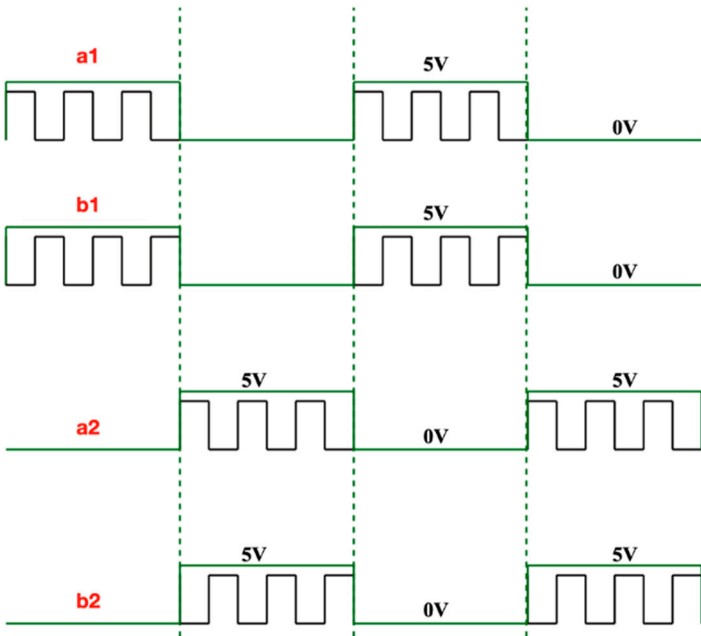

**Figure 40.** Modulation waveforms for the generation of the transformer input pulses. The desired output waveform (in green) is modulated with on–off keying.

In a practical implementation, the coupling factor $k$ of the transformers is poor at around $k \approx 0.5$. The coupling factor will reduce for an increasing number of switches (increasing primary winding length). For each transformer, the coupling of the primary wire to the toroidal cores will also vary slightly due to geometrical differences. Therefore, a compensation capacitor $C_p$ should be connected in series with the primary wire to compensate for the leakage inductance $L_k$. This increases the transformer efficiency and allows the transformer to be operated at high frequency in the resonance mode.

The modulation waveforms ($a_1$, $b_1$, $a_2$, and $b_2$) created by the micro-controller, as depicted in Figure 40, will be delivered to the low-voltage H-bridges of Figure 39 (left) to convert these unipolar waveforms into bipolar waveforms with the amplitude of 24 V. After that, these resulted waveforms will become complementary, as shown in Figure 41, and are filtered to recover the desired gate waveform and applied to the auxiliary switches.

The modulated waveforms applied to primary windings 1 and 2 are complementary, meaning that pulses will be present on $W_{1(s)}$ or $W_{2(s)}$, but not both simultaneously. When pulses are present on $W_{1(s)}$, these are rectified and sent to $Q_2$, which forces the HV MOSFET $Q_3$ to be in the off-state by pulling its gate low. On the other hand, when pulses are present on $W_{2(s)}$, these force $Q_2$ to be off and simultaneously deliver charge to the gate capacitor $C_{iss}$ to turn on $Q_3$. This allows for synchronous switching of the HV MOSFETs with only a minor impact of the parasitic properties of the circuit.

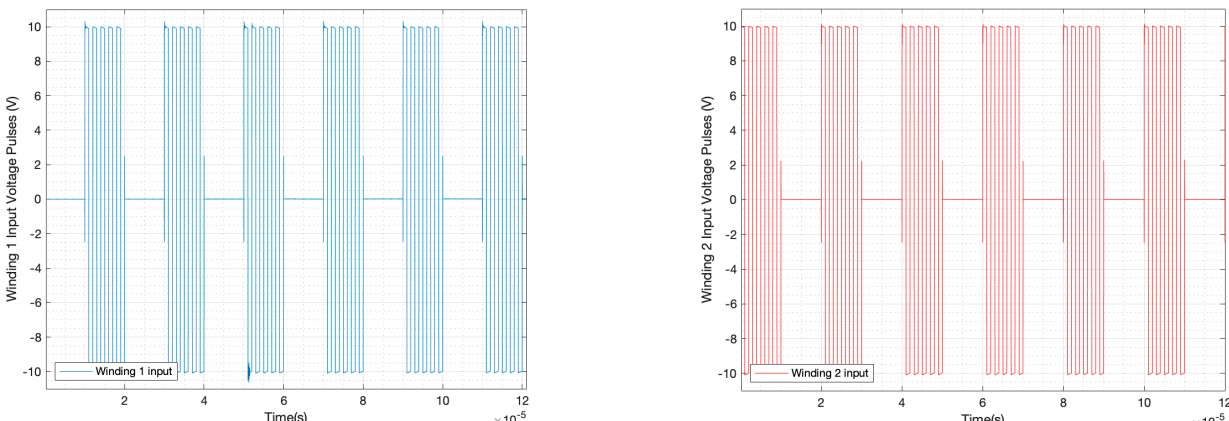

**Figure 41.** The desired bipolar and complementary modulation waveforms (gate-coupled transformer input pulses).

*LTspice Simulations of the Magnetically Isolated HV Gate Driver*

Figure 42 illustrates the schematic of the magnetically isolated gate driver with three series-connected MOSFETs in LTspice. The two gate-coupled transformers are utilized to trigger the three series-connected SiC MOSFETs with complementary bipolar gate pulses.

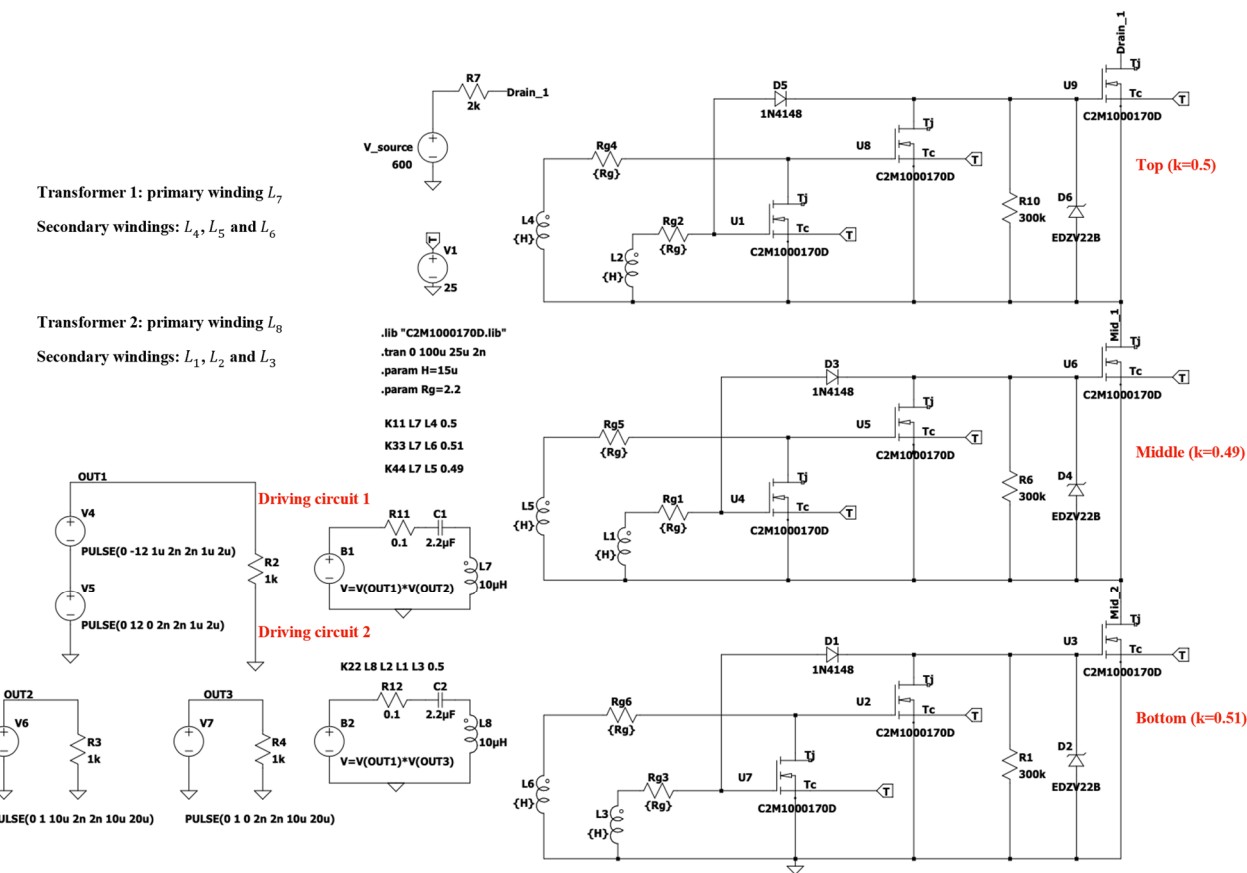

**Figure 42.** Schematic of the magnetically isolated HV gate driver based three series-connected SiC MOSFETs.

In the LTspice simulation model, $L_7$ denotes the primary windings, and $L_4$, $L_5$, and $L_6$ are the secondary windings of transformer $W_1$; $L_8$ denotes the primary windings, and $L_1$, $L_2$, and $L_3$ are the secondary windings of $W_2$. For each transformer, the driving pulses will be sent to its primary winding and then transferred to the secondary windings synchronously.

These driving pulses in the secondary windings control the auxiliary gate-side power switches ($U_1$, $U_4$, $U_7$ and $U_2$, $U_5$, $U_8$) for switching the main MOSFETs. As shown in Figure 42, the coupling factors of the windings corresponding to different main MOSFETs are not identical (e.g., $k_{top} = 0.5$, $k_{middle} = 0.51$, $k_{middle} = 0.49$). This non-ideality results in a slight imbalance in the drain-source voltages.

Consequently, the magnetically isolated gate driver-based MOSFET string requires fewer components than other balancing methods, and therefore a compact HV switch size can be achieved. Moreover, the scalability of this method is high since only two transformers with an arbitrary number of secondaries are required. The main disadvantage of this method is the poor coupling factor *k* of the transformers, which may generate some losses. However, applying a compensation capacitor at the primary side of the transformer can solve this issue. As shown in Figure 43, good $V_{DS}$ balancing can still be achieved with a poor transformer coupling factor. The experimental results based on this type of gate driver will be presented in a future paper [47].

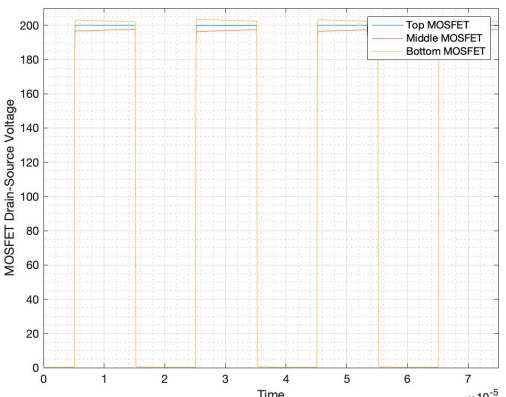 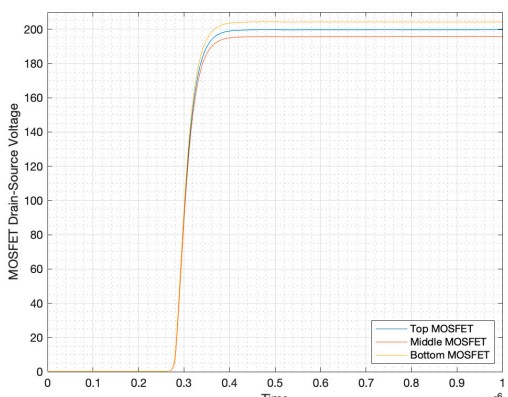

**Figure 43.** $V_{DS}$ overall waveforms (**left**) and $V_{DS}$ during turn-off period (**right**) of the magnetically isolated gate driver based SiC MOSFET string.

Similar to the optically isolated HV gate driver, other balancing methods could be applied to further improve voltage sharing among the MOSFET strings. However, the parameter variation ($\delta t_{d(on)}$, $\delta t_{d(off)}$, and $\delta C_{iss}$) of the circuit components is usually too low to generate a voltage imbalance on a level that would be destructive to the MOSFETs.

## 8. Conclusions

In this research, four techniques to ensure balanced $V_{DS}$ of the series-connected SiC MOSFETs were discussed. Satisfactory static balancing is achieved using balancing resistors at the cost of additional static power dissipation. The following methods were evaluated to achieve dynamic balancing during the switching transients: (i) the gate-balancing core method, (ii) improved RC snubbers, and (iii) the Zener clamping method. For HV applications, the gate-balancing core method proves to be a practical solution, offering good output performance. It ensures a short $V_{DS}$ rise time and excellent $V_{DS}$ sharing, even in the presence of a considerable $\delta t_{d(off)}$. The GBC method and the improved RC snubber method (b) can perform well in relatively low-voltage applications owing to their excellent voltage-balancing capability. However, it must be noted that improved RC snubber methods face challenges at higher input voltage levels due to the insulation requirements on the balancing transformer, which must withstand the entire stage voltage.

Furthermore, the research evaluated two types of high-voltage gate drivers and their ability to maintain balanced voltage sharing, even at high input voltage levels. The suitability of the techniques was assessed through experiments and LTspice simulations. For series-connected SiC MOSFETs with an input voltage lower than 8 kV, the optically isolated gate drivers discussed in Section 6 are recommended due to their lower propagation delay and ease of implementation. If the delay times of the gate driver components are not

matched, one of the $V_{DS}$ balancing techniques should be used in conjunction with the HV gate driver to ensure balanced drain-source voltage sharing. For input voltages exceeding 8 kV, the magnetically isolated gate driver demonstrated in Section 7 is recommended due to its small size and lower material cost. This is mainly because the gate coupling transformers can easily be extended to such voltage levels.

To further simplify the SiC MOSFET string and ensure balanced static and dynamic drain-source voltage sharing, careful sorting, binning, and matching of the SiC MOSFETs, optocouplers, non-isolated gate drivers, and other passive components could be undertaken prior to assembly. If executed correctly, this process could eliminate the need for $V_{DS}$ balancing, leading to a reduction in size and cost of the series-connected MOSFETs. Finally, it is recommended to cast the series-connected SiC MOSFETs in epoxy resin to prevent the occurrence of partial discharges that may otherwise occur at various locations in the circuit.

**Author Contributions:** Methodology, experiments, validations, LTspice simulations, investigation, visualization, reference collection, writing—review, draft preparation, and editing, W.Z.; methodology, investigation, visualization, editing, S.G.; methodology, reference collection, draft preparation, writing—review and editing, G.W.L.; methodology, supervision, investigation, writing—review and editing, M.G.N.; supervision, writing—review and editing, G.R. and P.V. All authors have read and agreed to the published version of the manuscript.

**Funding:** This research was funded by TKI Urban Energy grant number 1821403, which is highly appreciated.

**Data Availability Statement:** Data are contained within the article.

**Conflicts of Interest:** Gijs Lagerweij was employed by Prodrive Technologies. The remaining authors declare that the research was conducted in the absence of any commercial or financial relationships that could be construed as a potential conflict of interest.

## Abbreviations

The following abbreviations are used in this manuscript:

| | |
|---|---|
| GBC | Gate-balancing core |
| HF | High frequency |
| HV | High voltage |
| IC | Integrated circuit |
| IGBT | Insulated-gate bipolar transistor |
| MMC | Modular multilevel converter |
| MOSFET | Metal–oxide–semiconductor field-effect transistor |
| MV | Medium voltage |
| PE | Power electronic |
| PWM | Pulse-width modulation |
| Si | Silicon |
| SiC | Silicon carbide |
| SST | Solid-state transformer |

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
