# Peer review of "Comprehensive Investigation of Promising Techniques to Enhance the Voltage Sharing among SiC MOSFET Strings, Supported by Experimental and Simulation Validations"

_electronics, doi:10.3390/electronics13081481_

Round 1

Reviewer 1 Report

Comments and Suggestions for Authors

This is a good review plus experiment paper. The author could consider to add more reference in the introduction sections.

Reviewer 2 Report

Comments and Suggestions for Authors

Zhao et. al. reviewed several techniques to address the issue of unbalanced voltage sharing of series-connected MOSFETs in this paper and they investigated the effectiveness of every method. The paper is thorough and generally well written. The scientific method is reliable and the results are interesting to specific audience. Therefore, I in principle recommend publication. Nevertheless, I have the following comments/doubts, which I would appreciate if the authors can improve.

1. In the abstract, the authors mentioned "high-frequency (HF)". What is the frequency range?

2. The authors discussed certain circuits being superior to others depending on the voltage range of the HV. For example, they claimed "If the input voltage is higher than 8 kV, the magnetically isolated gate driver demonstrated in chapter 7 should be used". Nevertheless, in chapter 7 they did not do any experiment using HV over 8kV, is that correct? How can they make several claims (including this one) throughout the paper about which circuit to use in which voltage range, had they not done any voltage dependent measurement?

3. Figs. 3, 5, 10 and so on seems to be screenshots, which gives an unprofessional look. Can the authors re-plot them (like in Fig. 12)?

Reviewer 3 Report

Comments and Suggestions for Authors

The authors in this work, provide a detailed and concise evaluation of various approaches dealing with the unbalanced voltage sharing among the SiC MOSFET string. The pros and cons for each method are discussed and the suitable conditions for the application of each method is provided.          

The work is interesting and could be considered as a survey paper. The following comments should be incorporated in the revised version of the manuscript:

- Firstly, a thorough proofread of the manuscript is needed to rectify some existing typos/grammatical mistakes in the revised version.

-The manuscript seems to be written as a thesis. The authors used the word, “on this chapter, in this section,…”. So, the manuscript should be rewritten as a research paper and please avoid any redundancy.

-I wonder, what is the reference of the used equations in the manuscript, (like equation 1)?

-Moreover, any abbreviations should be descripted and do not assume the reader is familiar with them.

-Sometimes, the authors used experimental validation then simulation validation and analytical analysis. All the results should be plotted on the same graph.

-The resolution of some figures should be enhancement to be readable.

Comments on the Quality of English Language

A thorough proofread of the manuscript is needed to rectify some existing typos/grammatical mistakes in the revised version.

Round 2

Reviewer 3 Report

Comments and Suggestions for Authors

I would like to thank the authors for fulfilling the referees comments.